# Divergent trajectories to structural diversity impact patient survival in high grade serous ovarian cancer

Ailith Ewing [1,2] ✉, Alison Meynert[1], Ryan Silk [1], Stuart Aitken [1], Devin P. Bendixsen [1], Michael Churchman[3], Stuart L. Brown[1], Alhafidz Hamdan [1,2,4], Joanne Mattocks[1], Graeme R. Grimes[1], Tracy Ballinger[1], Robert L. Hollis [3], C. Simon Herrington [3,5], John P. Thomson[2,3], Kitty Sherwood [1,2,6], Thomas Parry[1,3], Edward Esiri-Bloom [1], Clare Bartos[3], Ian Croy[3], Michelle Ferguson[7,8], Mairi Lennie[9], Trevor McGoldrick[10], Neil McPhail[11], Nadeem Siddiqui[12], Rosalind Glasspool [13,14], Melanie Mackean[4], Fiona Nussey[4], Brian McDade[14], Darren Ennis [14,15], The Scottish Genomes Partnership*, Lynn McMahon[16], Athena Matakidou[17], Brian Dougherty[18], Ruth March[19], J. Carl Barrett[18], Iain A. McNeish [13,14,15], Andrew V. Biankin [14,20,21], Patricia Roxburgh [13,14,22], Charlie Gourley [3,22] & Colin A. Semple [1,22]

Deciphering the structural variation across tumour genomes is crucial to determine the events driving tumour progression and better understand tumour adaptation and evolution. High grade serous ovarian cancer (HGSOC) is an exemplar tumour type showing extreme, but poorly characterised structural diversity. Here, we comprehensively describe the mutational landscape driving HGSOC, exploiting a large (N = 324), deeply whole genome sequenced dataset. We reveal two divergent evolutionary trajectories, affecting patient survival and involving differing genomic environments. One involves homologous recombination repair deficiency (HRD) while the other is dominated by whole genome duplication (WGD) with frequent chromothripsis, breakage-fusion-bridges and extra-chromosomal DNA. These trajectories contribute to structural variation hotspots, containing candidate driver genes with significantly altered expression. While structural variation predominantly drives tumorigenesis, we find high mtDNA mutation loads associated with shorter patient survival. We show that a combination of mutations in the mitochondrial and nuclear genomes impact prognosis, suggesting strategies for patient stratification.

High-grade serous ovarian cancer (HGSOC) is the most common type of ovarian cancer. It usually presents with disease that has spread beyond the pelvis, and while initially sensitive to platinum-containing chemotherapy in 70–80% of cases[1], historically over 80% of cases relapse with a median overall survival of less than 5 years[2,3]. The advent of first-line maintenance PARP inhibitor therapy is improving survival, particularly in patients whose tumours are homologous recombination repair deficient (HRD)[4]. However, around 50% of tumours are not

A full list of affiliations appears at the end of the paper. *A list of authors and their affiliations appears at the end of the paper. ✉e-mail: ailith.ewing@ed.ac.uk

HRD[5] and the molecular drivers and therapeutic vulnerabilities in this patient subset with poorer prognosis are much less well characterised.

Recent studies of the HGSOC mutational landscape have noted the problems caused by structural complexity at many loci, potentially obscuring driver events and useful biomarkers[5]. The genomic aberrations at these repeatedly mutated loci require whole-genome sequencing (WGS) data to be accurately resolved, and their combined impact determined. However, recurrent aberrations at a number of genes have been repeatedly reported. Initial analyses of the HGSOC genome using exome sequencing noted recurrence of copy number alterations (CNAs) and predicted driver variants in genes including: *TP53*, *NF1*, *RB1*, *CDK12*, *BRCA1* and *BRCA2*[6,7]. An early WGS study (*N* = 80) recovered a similar list of recurrently disrupted genes, and also reported amplification of the *CCNE1* oncogene in 19% of samples[8]. However, a more recent WGS study (*N* = 118) identified less frequent driver alterations impacting additional genes, and reported recurrent deletions−rather than amplifications−at *CCNE1* in 45% of patients[9]. The same study also suggested that the majority of driver events in HGSOC are likely to be mediated by somatic structural variants (SVs) and copy number alterations (CNAs). Thus, the driver landscape in HGSOC remains controversial, highlighting the need for comprehensive analyses of the mutational complexity of HGSOC genomes[10] and elucidation of its clinical impact in larger WGS cohorts.

HGSOC is subject to catastrophic mutational events, generating whole genome duplication (WGD)[11,12], complex structural variants (cSV) such as chromothripsis or 'chromosome shattering'[13], the production of extrachromosomal circular DNA (ecDNA)[14] and other cSV types involving overlapping amplifications and inversions[15,16]. However, the interdependencies between cSV types have not been studied in detail in any tumour type, including HGSOC, and their individual and joint impacts on patient survival remain poorly understood[17]. HGSOC tumour cells are also known to possess particularly abundant mitochondria carrying frequent somatic mtDNA mutations, though the best powered studies to date have failed to find associations between mitochondrial perturbation and patient outcomes[18–20]. Epistatic interactions between somatic mutations, across different scales of size and complexity, are thought to emerge frequently between different driver mutations during tumorigenesis[21,22], but remain poorly studied. Several previous studies have used deep WGS to characterise HGSOC tumour samples[6,8,23–25] combined with gene expression profiling and other technologies, but have been constrained by modest sample sizes with limited power to detect recurrent mutations and any epistatic interactions between them.

Here we combine newly generated WGS and gene expression data from these tumours with uniformly processed data from previously published studies[6,8,23,25], to construct a large WGS HGSOC cohort with matched RNA-seq (WGS *N* = 324). We comprehensively describe the mutational landscape of HGSOC to define candidate driver genes subject to recurrent somatic short nucleotide variants (SNV), SV or CNA and determine the diversity and interactions of mutations driving tumorigenesis. We reveal the divergent evolutionary trajectories adopted by different HGSOC tumours to generate structural diversity and gain insights into the interdependencies of cSV types. We also perform a comprehensive study of mtDNA mutations in HGSOC, revealing their association with patient survival. Finally, we construct an overarching model for HGSOC prognosis based upon all features of the mutational landscape and identify the most influential features.

## Results

### Extreme structural diversity generates collateral damage across the HGSOC genome

The combined cohort (*N* = 324) data is composed of deep WGS (median 71X coverage; range: 52–136×) and RNA-seq data from primary HGSOC tumours. WGS data from matched tumour and normal blood samples, and RNA-seq data from the same tumours, were uniformly remapped and analysed to generate a harmonised dataset of consensus somatic mutation calls and gene expression for five HGSOC WGS sub-cohorts: Scottish High Grade Serous Ovarian Cohort (SHGSOC, *N* = 115[24]), Australian Ovarian Cancer Study (AOCS, *N* = 80[8]), British Colombia Cancer Agency (BCCA, *N* = 59[23]), The Cancer Genome Atlas (TCGA, *N* = 44[6]), and MD Anderson (MDA, *N* = 26[25]). Although each sub-cohort was independently ascertained and sequenced, uniform processing and analysis revealed consistent mutational landscapes (Fig. 1; Supplementary Fig. 1). These landscapes are dominated by somatic SVs and CNAs occurring with similar rates and genomic spans (Supplementary Fig. 1) rather than by SNVs. SVs and CNAs are dominated by large duplications, with duplications composing 89% of all CNAs (Fig. 1), consistent with previous studies[26]. The compound effects of SNV, SV and CNA are predicted to disrupt the function of several thousand genes in each tumour (Fig. 1B), representing a large predicted deleterious mutation load. However, with the exception of *TP53*, the most recurrently affected genes are impacted by likely passenger rather than driver events (Fig. 1C), appearing to suffer collateral damage due to their proximity to SV/CNA hotspots. For example, the top 10% of most heavily disrupted genes (*N* = 1932) are only modestly enriched for genes with known roles in cancer (Cancer Gene Census (CGC)[27])(OR: 1.5; chi-squared *p* = 4.4 × 10⁻⁴). As expected, samples also show enrichment of low-frequency germline variants known to increase the risk of HGSOC (Supplementary Data 3).

These events occur on a background of frequent HRD (56% of samples) and WGD (49%). HRD and WGD are not mutually exclusive but are anti-correlated, such that HRD is depleted in samples with WGD (odds ratio (OR) 0.56; chi-squared *p* = 0.015). HRD tumours are significantly enriched for deletions (Wilcoxon *p*-value < 2.8 × 10⁻⁵) and WGD samples are significantly enriched for duplications (Wilcoxon *p*-value < 1.8 × 10⁻⁸) as expected (Fig. 2A, Supplementary Fig. 2). The high frequency of CNAs provides insight into the underlying processes generating structural diversity, with dominant contributions of CNAs linked with HRD and chromosome mis-segregation, as reflected in the presence of known copy number signatures[28] as expected based on previous reports (Supplementary Fig. 3; Supplementary Data 4). A variety of complex SVs (cSVs) are also highly prevalent in this tumour type (Fig. 1A; Supplementary Data 1; Suppl. Fig. 4), particularly chromoplexy (55% of samples), chromothripsis (40%), pyrgo (28%) and breakage-fusion-bridge events (BFB) (27%). Diverse ecDNA species were observed in a minority (16%) of tumours across sub-cohorts, and in 8% of tumours the predicted ecDNA structures contained an amplified oncogene (Supplementary Data 2). The only major difference in cSV occurrence between sub-cohorts was a strong enrichment (OR: 9.28; chi-squared *p* = 4.1 × 10⁻¹³) of ecDNA in AOCS samples, which are enriched for chemoresistant tumours.

### Oncogene amplification mediated by multiple classes of complex variants

Both SVs and CNAs were non-randomly distributed across chromosomes in this cohort (Fig. 2A; Supplementary Fig. 5). We observed notable enrichments of all SV classes on chromosomes 8, 11, 12, 19 and 20 (Methods; Supplementary Fig. 5E; Supplementary Data 5). Chromosome 19 also represents a previously unknown hotspot for cSV in HGSOC as it is significantly enriched for BFB, chromothripsis, ecDNA and chromoplexy (Supplementary Fig. 5F; Supplementary Data 5). Remarkably, this hotspot was seen across sub-cohorts, supporting the highly rearranged state of this chromosome as a general feature of HGSOC (Supplementary Data 5). The densities of SVs (including when restricted to simple SVs only), chromothripsis and BFB on chromosome 19 peak at *CCNE1* (Fig. 2C), a known HGSOC oncogene subject to recurrent amplification[5,29]. Amplification of *CCNE1* has been associated with higher genome-wide rates of large tandem duplication[30], which we also observe here. Discriminating between possible mechanisms underlying a genomic region subject to complex rearrangement

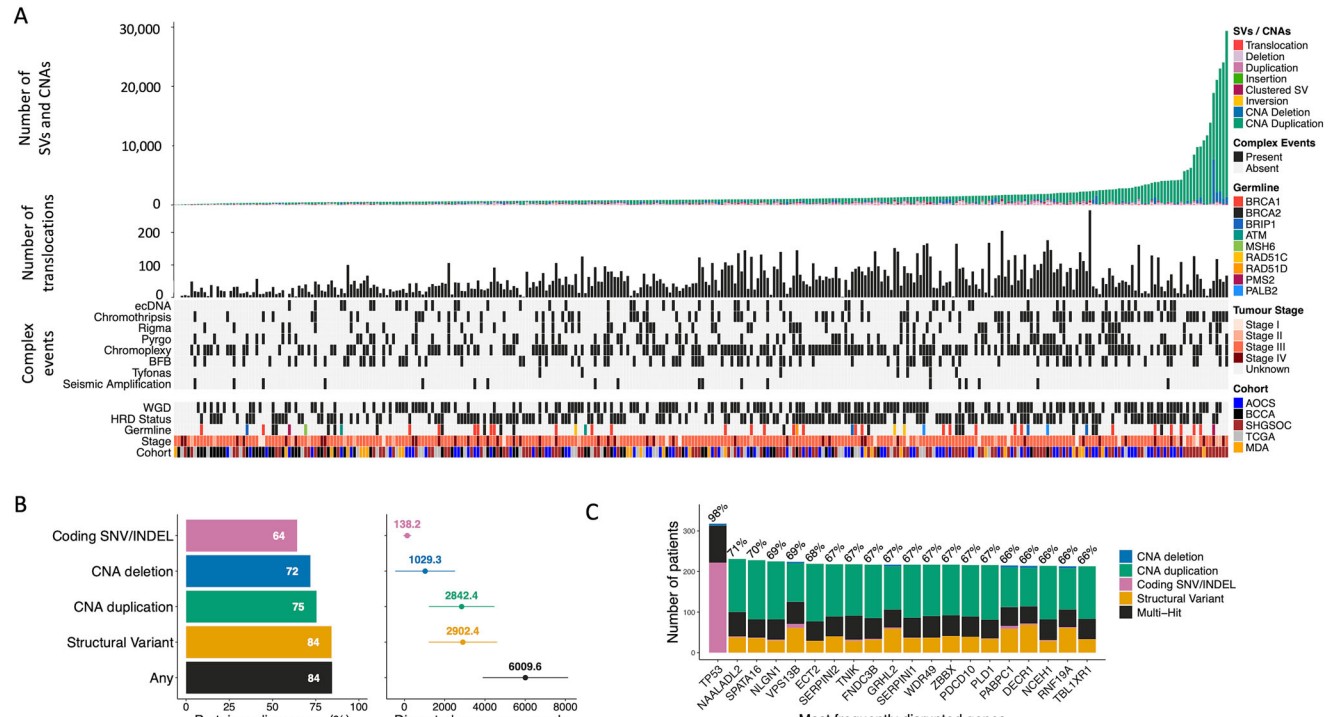

**Fig. 1 | Structural complexity in HGSOC. A** Structural variants are abundant across the combined tumour cohort (*N* = 324), reflected in the frequencies of SV and CNA calls and particularly of translocations and complex events. Samples also show frequent HRD and WGD across tumour stages and cohorts (Supplementary Data 1). Pathogenic germline variants are relatively enriched in known HGSOC susceptibility genes. **B** Of all protein-coding genes in the genome (*N* = 19,329), most are impacted by each class of variation at the cohort level (bar chart on left) but the numbers of disrupted genes per sample (forest plot on right; mean ± SD) are dominated by SVs and CNAs and impact a smaller number of genes per sample.

Variants predicted to disrupt function are nonsynonymous SNVs of high/moderate impact and SVs or CNAs overlapping ≥1 protein-coding exon. **C** The most frequently disrupted genes overall are not enriched for known cancer genes, tend to be longer than average and to intersect recurrent CNA hotspots. The most frequently disrupted gene is TP53, where there is near ubiquitous mutation. NAA-LADL2 is an unusually long (1.37 Mb) gene and is located within a common fragile site frequently altered across many tumour types[26]. (All somatic variant calls predicted to disrupt protein-coding genes are provided in Source Data.).

remains challenging, and 298/776 (38%) of the complex SVs identified overlap with at least one other type of SV. In such cases, we have included both complex SV calls throughout; however, if we were to exclude these ambiguously rearranged regions, we continue to observe a peak of breakage fusion bridges, chromothripsis and pyrgo at *CCNE1* and uniquely chromothripsis or uniquely ecDNA regions also remain significantly enriched on chromosome 19. Samples with BFB events at this locus acquire higher *CCNE1* copy number amplifications than those with simple SV duplications (Supplementary Fig. 6). *CCNE1* expression is also higher in the presence of SV duplications of any type and particularly so in the tumours where ecDNA and BFBs co-occur (fold change relative to simple SV duplication: 2.8 (95% CI: 1.7–4.6), adj. *p*-value = 0.05) (Supplementary Data 6). This capability to amplify oncogenes beyond what is feasible via simple duplication events is likely to be advantageous to an evolving tumour. Similar effects as a result of ecDNA-mediated amplification have been reported in other cancer types and are posited to lead to treatment resistance and subsequent poorer prognosis[31].

We discovered additional genomic regions with significant enrichment[32] of SVs–SV hotspots–at a number of loci across the genome (FDR < 0.05; Supplementary Data 7). In addition to the strongest signal across all SV types observed on chromosome 19, there are deletion and inversion hotspots on chromosome 2, a further inversion hotspot on chromosome 10 and multiple further loci enriched for SV breakpoints regardless of type throughout the genome. Moreover, we observe significant hotspots of breakpoints driven exclusively by translocations. We also determined CNA hotspots across the cohort based upon significant enrichment[33] of variants in regions varying in size from tens of kb up to multi-megabase regions

encompassing entire chromosome arms (Methods). Of 44 focal CNA hotspots examined, 25 were enriched for deletions and 19 for amplifications (Fig. 2A, Supplementary Data 8). The proportion of samples with CNAs at a given CNA hotspot varied widely from 4% to 88%, with HRD and WGD tumours accumulating higher CNA loads at deletion and amplification hotspots respectively (Fig. 2B). Although many CNA hotspot regions include genes with known roles in cancer[27], they occur approximately in the numbers expected by chance (deletion hotspots OR = 1.2, CI: 0.88–1.7 chi-squared *p* = 0.18; duplication hotspots OR = 1.4, CI: 0.99–1.96 *p* = 0.043). A critical question is therefore whether the expression patterns of such genes are altered in response to the CNA burdens they incur, to affect the tumour phenotype.

## Hotspots of structural alteration implicate candidate driver genes

We devised an approach to interrogate genes within CNA hotspots, exploiting the matched expression data available for the sub-cohorts making up the combined cohort, to rigorously prioritise candidate drivers. We assume that CNA-associated driver genes should show significant alterations in expression, consistent with the CNAs that impact them. For each cancer gene census gene present in a CNA hotspot (449 genes in total) we calculated the differential expression (DE) seen between samples with high copy number versus those with low copy number in each sub-cohort, and the associated false discovery rate (FDR) for DE genes seen across multiple sub-cohorts (Methods; Supplementary Data 9). Given the likely presence of confounding variation in the expression data (reflecting cellular heterogeneity and technical variation), these tests are necessarily conservative. Supporting this, we found that significantly lower

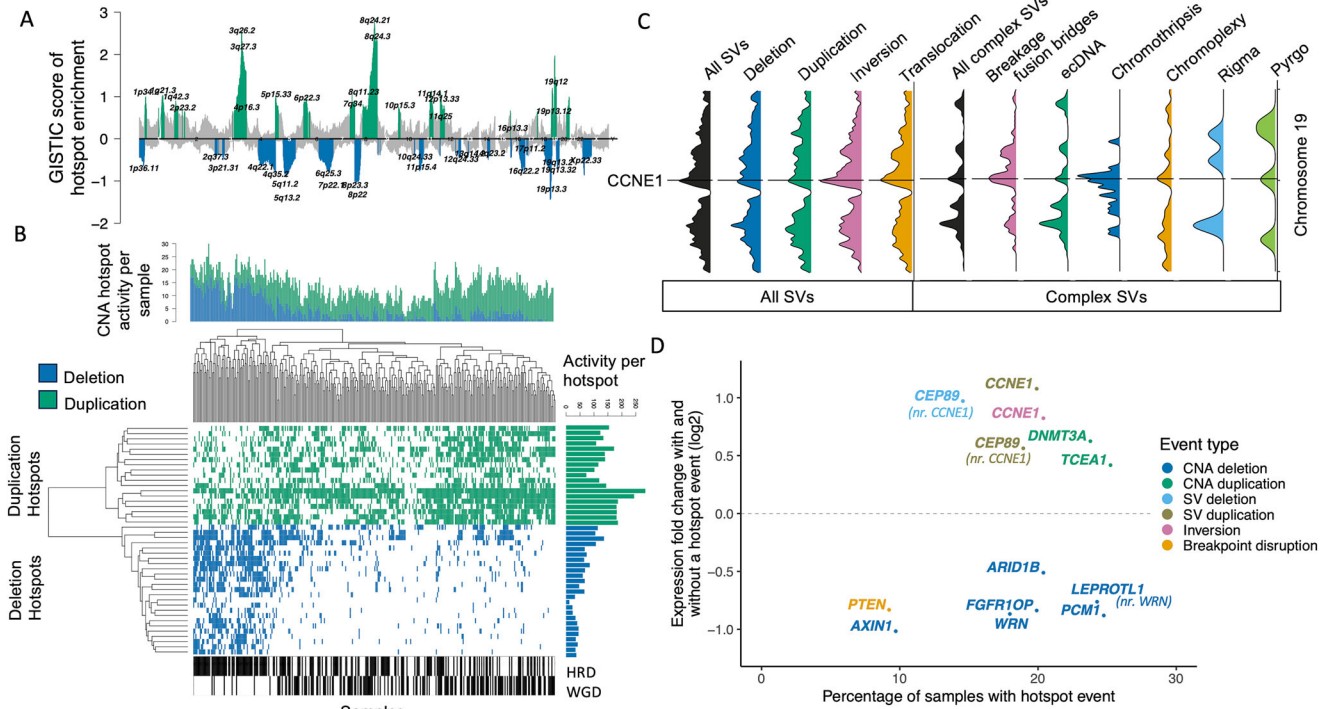

**Fig. 2 | CNAs and SVs form hotspots throughout the genome, which reveal candidate driver genes. A** GISTIC enrichment analysis reveals CNA hotspots. Duplication peaks are green, and deletion peaks are blue. **B** CNA hotspot activity varies across samples, with greater activity at deletion hotspots in HRD samples and greater duplication activity in WGD samples. **C** SVs are enriched on chromosome 19 and peak at CCNE1. In particular, there is a relative excess of inversions, translocations, breakage-fusion bridges and chromothripsis implicating this gene. **D** Cancer gene census genes in SV/CNA hotspots that are differently expressed across samples in the presence of the event type driving the hotspot. Genes in CNA deletion hotspots or hotspots of breakpoint enrichment have significantly lower expression, while genes in duplication (SV or CNA) or inversion hotspots have significantly higher expression. CEP89 SV deletions and SV duplications are both associated with increased expression, reflecting the high level of overlapping genomic instability occurring at this locus, which includes CCNE1. CNA deletions at LEPROTL1 are also associated with reduced expression, but this likely represents the same signal as that observed at nearby WRN. Expression fold changes are log2 transformed and are robust to the percentage of samples with the hotspot event. SV and CNA hotspot data are in Supplementary Data 7 and 8, respectively.

expression of *PTEN* (a known tumour suppressor gene (TSG) in HGSOC) was associated with CNA deletion, but as this was seen in only one sub-cohort, it failed to reach significance (FDR < 0.05) and was excluded. Eight genes were identified as candidate drivers with several neighbouring genes altered simultaneously (Fig. 2D). The expression of *AXIN1* (16p deletion, lower expression), *DNMT3A* (2q duplication, higher expression) and *TCEA1* (11q duplication, higher expression) all reflected the CNA burdens observed. Significantly lower expression of *PCM1, WRN* and *LEPROTL1* was associated with deletion, and all three are located within the same 8p deletion hotspot. Similarly, *ARID1B* and *FGFR1OP* are both within a 6q deletion hotspot, and both show lower expression in response to CNA deletions. Notably, the 8p deletion hotspot has been reported in many other tumour types[34] and may confer multiple advantageous traits on tumours[35]. The genes underlying these effects were unknown, though recent work has shown that *WRN* deletion increases cell growth in vitro, suggesting *WRN* is the causal haploinsufficient pan-cancer TSG underlying the 8p deletion hotspot and that *PCM1* and *LEPROTL1* are likely co-deleted with WRN rather than driver genes themselves[36].

An analogous approach was taken to prioritise candidate genes based upon FDR-corrected differential expression in SV hotspots (Fig. 2D; Supplementary Data 10). Two genes emerged as SV-associated driver candidates: significantly lower expression of *PTEN* (a known TSG in HGSOC) was associated with disruption by SV breakpoints, and higher expression of the HGSOC oncogene *CCNE1* was associated with both duplications and inversions (including fold-back inversions where an inversion co-occurs with a duplication). The *CEP89* gene neighbouring *CCNE1* also passed the FDR threshold as a

driver candidate, but was associated with both duplications and deletions, suggesting it may simply be a marker for the complex disruptions accumulating at the *CCNE1* locus rather than a driver in its own right. Differentially expressed gene functions and pathways associated with the presence of candidate CNA/SV driver genes are detectable (Supplementary Data 16 and 17), but should be interpreted with caution, due to the substantial cellular heterogeneity and potential presence of confounding mutations in HGSOC tumour samples. Genes and noncoding regions carrying recurrent SNVs were analysed using multiple driver prediction algorithms (Supplementary Fig. 7; Supplementary Data 11) and recovered 6 known HGSOC driver genes (*TP53, NF1, BRCA1, CDK12, BRCA2, RB1*), plus another 2 genes with multiple paralogues: *SLC35G5* which has been reported previously as a source of artefacts[37] and *TAS2R43*. Many of the other recurrently mutated genes have been identified as likely false positives in previous studies[38,39] and in common with a recent study[9], we found no convincing evidence of SNV drivers in noncoding regions (Supplementary Fig. 7).

The combined driver landscape—encompassing genes driven by SNVs, SVs and CNAs—is dominated by diverse structural alterations (Fig. 3). The predicted pathogenic alteration rates (Fig. 3A) for known HGSOC genes are higher than seen in previous studies lacking WGS data, suggesting improved diagnostic power for WGS in structurally diverse tumours relative to exome or panel sequencing. Disruption of *NF1, PTEN* and *RB1* has been reported as recurrent events in HGSOC[5,40], but the structural complexity seen at these loci[8] may have obscured inactivating alterations in studies lacking WGS data. Our current data confirm that these tumour suppressors are frequently disrupted by

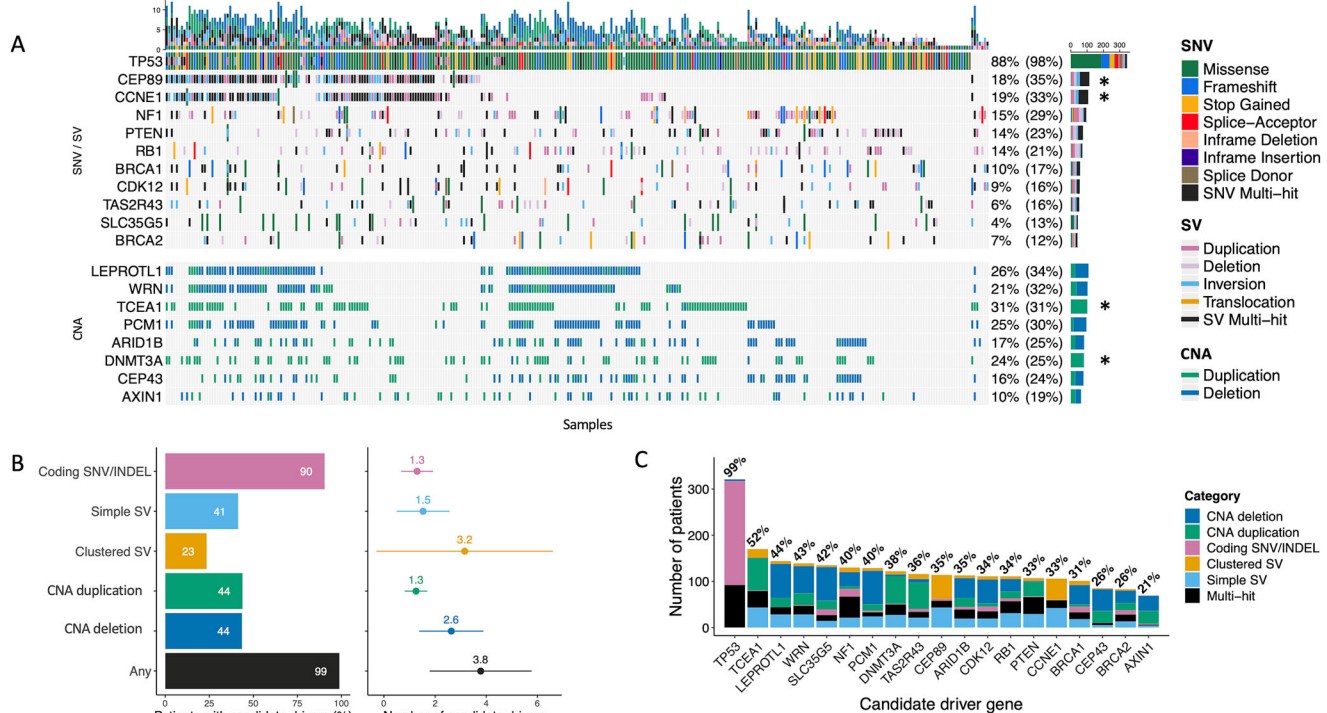

**Fig. 3 | Diverse somatic mutation classes underlie HGSOC candidate driver genes. A** Combined oncoplot indicates predicted driver genes (Methods) subject to recurrent pathogenic SNV/SV/CNA mutations predicted to impact function (SNVs annotated are nonsynonymous variants of HIGH or MODERATE impact by VEP; SVs or CNAs overlapping ≥1 exon). The unbracketed percentage is the percentage of patients with a predicted pathogenic SNV/SV or CNA. For SNVs/ SVs, this is either an SNV or a deletion, or in the case of the asterisked rows, representing genes associated with gain of function, it is an SNV or duplication. The bracketed percentage is less conservative and is the percentage of patients with any annotated event. The vertical bar plot represents the mutational burden of predicted deleterious SNVs/SVs. **B** The horizontal bar plot (left) represents the proportion of patients with different types of pathogenic driver mutations. The forest plot (right) represents the mean number of each type of driver mutation across tumours with at least one event and the standard deviation (whiskers), based on N = 324 patients. **C** Total mutation frequencies in candidate driver genes. The proportions of patients with somatic alterations of any kind in each gene, whether predicted to be pathogenic or not. Clustered SVs are members of SV breakpoint clusters from ClusterSV[26]; the Multi-Hit category represents patients with >1 somatic alteration in the gene. (All somatic variant calls predicted to disrupt protein-coding genes are provided in Source Data.).

structural alterations, with pathogenic genomic events of any kind seen in 15% (*NF1*), 14% (*PTEN*) and 14% (*RB1*) of samples. The overall rates of alteration to *NF1*, *PTEN* and *RB1* are similar, if conservative in the case of *NF1*, to those seen in another WGS project (N = 118) ascertaining combined SNV/SV/CNA loads[41], which reported alteration rates of 24%, 14% and 19%, respectively. Frequent pathogenic structural alterations to *BRCA1* and *BRCA2* are consistent with the role of SV/CNA mutations in HRD[24]. Overall, each HGSOC sample is predicted to contain 3.8 driver variants on average, with only 1.3 contributed by SNVs and the remainder involving SVs and CNAs (Fig. 3B), which is similar to recent estimates based on WGS data ([41]; HGSOC N = 118). In addition, many candidate driver SVs are members of significant SV clusters (Fig. 3B, C)[26], which often indicate cSV events such as chromothripsis, suggesting that these events may be acting as drivers in some contexts. HRD or WGD differ in their genome-wide burden of deletion and duplication, respectively, however, this appears to have no bearing on either the number or distribution of drivers by mutation type (Supplementary Fig. 8).

### HRD and WGD underlie different evolutionary trajectories to structural diversity

The chaotic HGSOC genome harbours frequent occurrences of most cSV types reported to date, presenting a disordered and, at first sight, uninterpretable picture. Using state-of-the-art algorithms, we have discovered high frequencies of known cSV types across the cohort, with chromoplexy, chromothripsis, pyrgo, rigma, BFB and ecDNA each seen in >10% of samples (Supplementary Data 1). This substantial WGS

cohort is sufficiently powered to reveal significant biases in the patterns of cSV co-occurrence, a unique opportunity to study their genomic distributions and interactions in detail in patient samples, bringing clarity to our understanding of a tangled landscape.

We define two evolutionary trajectories to complex structural diversity (Fig. 4). Each of these trajectories reflects the processes generating the cSV landscape in HGSOC and the underlying genomic state of the tumour. One trajectory involves HRD (Fig. 4B, purple) which is positively associated with chromoplexy, while the other involves WGD (Fig. 4B, green) and a strong tendency to the acquisition of other cSV types (Fig. 4B). Although these trajectories are not perfectly anti-correlated, it is evident that the divergent underlying genomic states of (i) deficiency in DNA repair and (ii) aneuploidy, are key aspects of HGSOC tumour biology which relate to different cSV profiles (Fig. 4B). Striking patterns of anti-correlation exist between HRD and all cSV types except chromoplexy (Fig. 4C). We observe positive correlations between different classes of cSV and this is driven in part by the ambiguity in discriminating between cSV types. Particularly strong associations are seen between the fraction of the tumour genome duplicated and two highly disruptive cSV types– chromothripsis and BFB (Fig. 4D). Abundant chromothripsis and BFB events account for disproportionate disruptions of genomic structure across the cohort, encompassing large fractions of the genome and causing many SV breakpoints in affected samples. The co-occurrence of these events with WGD suggests that WGD may buffer the particularly disruptive effects of these catastrophic events and limit their impacts on gene function.

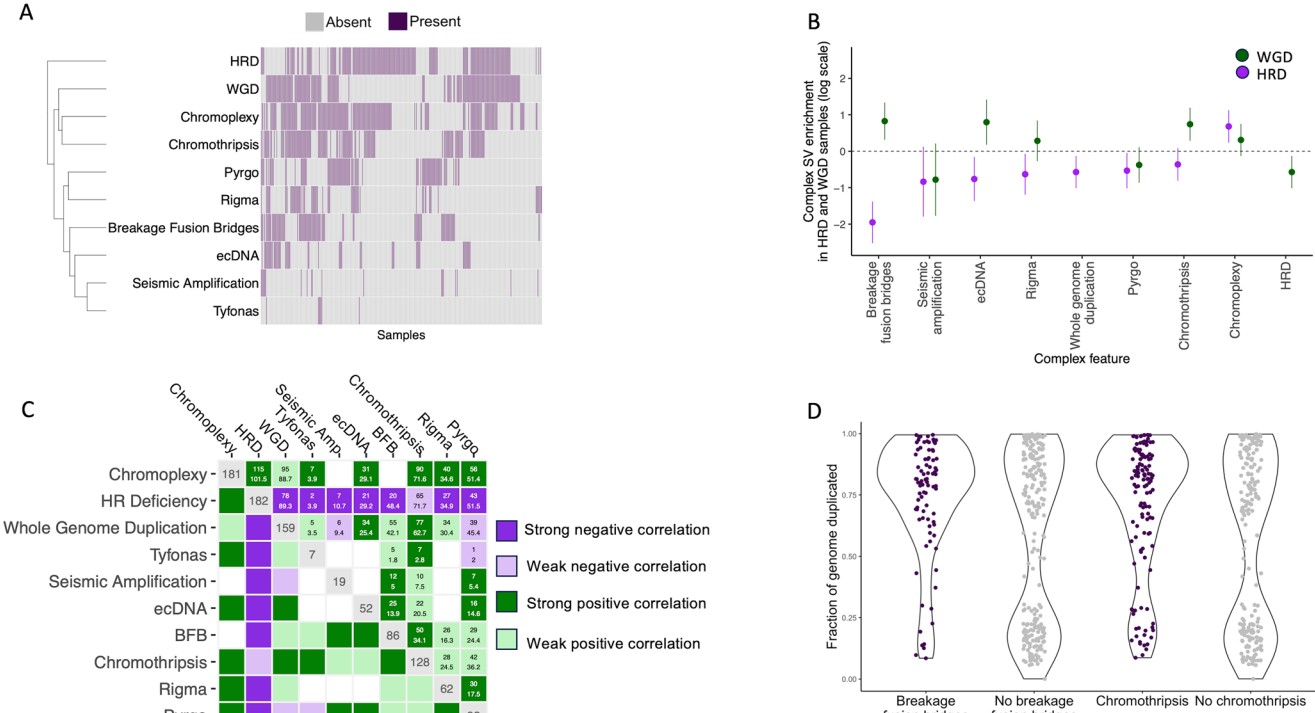

**Fig. 4 | Patterns of co-occurrence and mutual exclusivity of complex structural variant classes and their relationship with WGD and HRD. A** Abundance and co-occurrence of all complex events reveals two clusters of events defined by negative correlation between HRD and WGD and other cSV types. **B** Complex SVs are depleted in HRD samples (purple; $N = 181$) with the exception of chromoplexy. Breakage fusion bridge cycles, ecDNA and chromothripsis are enriched in whole genome duplicated samples ($N = 158$). Points represent log odds ratios, and lines represent 95% confidence intervals. **C** Biased co-occurrence and mutual exclusivity of cSV classes support divergent tumour evolutionary trajectories with significant association seen between HRD and chromoplexy, but exclusivity between HRD and all other features. Cell counts represent observed vs expected counts of event co-occurrence in samples. **D** Association between the presence of chromothripsis and breakage fusion bridges (x-axis) and the estimated fraction of the genome duplicated (y-axis) across all samples. Enrichments (two-sided Fisher's exact tests) in WGD samples of: chromothripsis (OR (95% CI) = 2.1 (1.3, 3.3); adj. p-value = $5.12 \times 10^{-3}$); and breakage fusion bridges (OR (95% CI) = 2.3 (1.4, 3.8); adj. p-value = $5.12 \times 10^{-3}$). HRD, WGD and complex SV data are in Supplementary Data 1.

Extensive characterisation of the HR proficient group of HGSOC is of great clinical importance as these patients have fewer options for targeted treatment, with patients with HRD tumours benefitting more from PARP inhibition. We observe that patients with a greater than average number of chromosomes (2) predicted to be implicated in chromothripsis in their tumour genomes—or severe chromothripsis—had better prognoses than those patients with fewer chromosomes involved (Supplementary Data 14). These severe events represent the most extreme levels of genomic instability occurring in a subset of cases ($N = 26$). Other cSV types, including ecDNA, showed weaker evidence for association with survival (Supplementary Data 12), contrary to previous pan-cancer reports[17] suggesting that the impact of ecDNA may differ in HGSOC from its impact in other tumour types.

**Mitochondrial and nuclear mutations have combinatorial impacts on patient survival**

Many tumour types, including HGSOC, are known to accumulate somatic SNVs in their mtDNA[18], but the consequences for mitochondrial (mt) function and patient survival remain unknown. We have found high mtDNA copy numbers and abundant somatic SNVs in tumour samples (Fig. 5A; Supplementary Fig. 9A; Supplementary Data 13), such that genes encoded in mtDNA suffer truncating and missense mutations at higher rates than in most other genes, including all known TSGs other than *TP53* (Supp Fig. 9B, C). The highest deleterious mutation loads accumulate at particular genes (Fig. 5B) and are predicted to disproportionately affect the function of mitochondrial Complex I (CI) and Complex IV (CIV) genes (Fig. 5C). Remarkably, the predicted deleterious SNV loads in mtDNA are also a potent biomarker

of poor patient prognosis, occurring in 26% ($N = 73$) of samples in the cohort with complete survival data ($N = 277$), with mutations of higher heteroplasmy showing the largest effects (Fig. 5D; Fig. 6E). Notably, no associations with survival were seen for synonymous SNVs or SNVs occurring in mitochondrial RNA genes (Fig. 5E; Supplementary Fig. 9D, E), demonstrating that the impact on patient survival is mediated via the compromised functions of protein-coding mitochondrial genes, particularly those in CI. Comparisons of gene expression between tumour groups with and without deleterious mtDNA variants show modest differences, with only 47 nuclear genes significantly differentially expressed (Supplementary Fig. S13A). Pathways enriched in these genes related to higher OXPHOS and reactive oxygen species production in tumours lacking variants, suggesting, as expected, that disruptive mtDNA variants may compromise normal mitochondrial function (Supplementary Fig. S13B).

Recent studies have reported deleterious somatic SNV loads in mitochondrial genes in renal, thyroid, and colorectal tumour types[20], but the associations of these loads with alterations to the nuclear genome are poorly studied. The rich mutational landscape of HGSOC described here provides an unusual opportunity to study these associations. Several trends emerge (Supplementary Fig. 10) using a Bayesian inference approach[22] to study co-occurrence patterns across all somatic alterations. Firstly, within the mitochondrial genome, there is an association between disrupted CI and CIII genes, suggesting specific alterations to mitochondrial metabolism. Secondly, these mitochondrial alterations significantly co-occur with WGD. Thirdly, this association appears to be attributable to WGD itself rather than the cSV (such as chromothripsis, BFB and ecDNA) that are correlated with

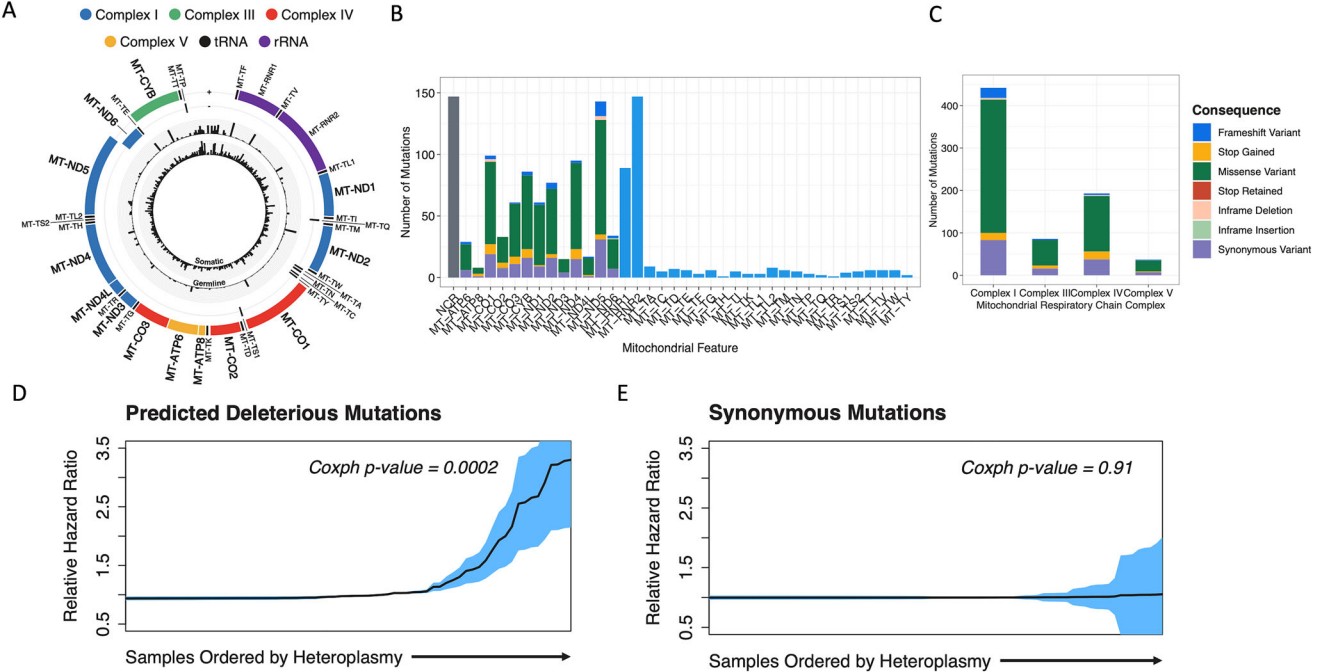

**Fig. 5 | Deleterious mtDNA mutation loads are a potent biomarker of overall survival. A** Somatic (inner ring) and germline (outer) SNV frequencies across the cohort in mitochondrial encoded genes (black: single nucleotide variants; red: indels). **B** Abundant somatic SNVs disproportionately impact protein-coding genes in mtDNA (758 SNVs in coding genes of 1245 SNVs in total). **C** SNVs categorised by VEP functional impact annotation include many protein-altering variants expected to alter mitochondrial complex functions. **D** Overall survival Cox proportional hazard (PH) ratio (95% confidence intervals in blue) increases with increasing heteroplasmy of deleterious mtDNA mutations (Cox PH $p$-value = 0.0002; $N$ = 277 patients, stratified by cohort and adjusted for age, stage and HRD status). **E** Overall survival Cox PH ratio (95% confidence intervals in blue) is stable with increasing heteroplasmy of synonymous mtDNA mutations (Cox PH $p$-value = 0.91; $N$ = 277 patients, stratified by cohort and adjusted for age, stage and HRD status). mtDNA SNV data are in Supplementary Data 13.

WGD (Supplementary Fig. 10A). In fact, the accumulation of somatic SNVs in most mitochondrial genes tends to be higher in the presence of WGD, and tends to be lower in tumours with HRD, suggesting less tolerance of disruptive mitochondrial DNA mutations in the presence of HRD (Supplementary Fig. 10B). These interdependencies raise the question of whether the effects of mitochondrial mutation loads on survival are independent of the known influences of HRD, WGD and the many other somatic alterations of the nuclear genome.

We examined the dominant mutational features of the nuclear and mitochondrial genomes identified here to identify the variables driving differential survival. We systematically examined the associations of all individual features with overall survival (OS) time after diagnosis, including the presence of genomic aberration in genes with demonstrable recurrent SNV/SV/CNA (Fig. 3), cSV types (Fig. 4), and deleterious mitochondrial SNVs (Fig. 5). Of these 34 binary features, we found that 8 were individually associated with overall survival using Cox proportional hazards models stratified by cohort (Fig. 6A; Supplementary Data 12). As expected, these 8 features included the presence of HRD and tumour FIGO stage, which have well-established effects on OS. Other individually significant features were *CDK12* SNVs, deleterious mt SNVs, and breakage-fusion-bridge genes. However, given the abundant interdependencies between these and other features (Supplementary Fig. 9A), we employed integrative modelling to estimate the independent effects of all 34 features. Analysis using regularised Cox proportional hazards regression with an elastic net penalty, stratifying by cohort, revealed a refined model with redundant features pruned (Fig. 6C; Supplementary Data 14). This model retained FIGO stage at diagnosis and HRD, as well as 6 other features, of which 2 were significantly associated (adjusted $p$ < 0.05) in the multivariable model with overall survival. *CDK12* SNVs (Fig. 6D) and deleterious mitochondrial SNVs (Fig. 6E) were associated with poorer prognosis, whereas the presence of severe chromothripsis (Fig. 6F) and *CEP89*

duplication, at the *CCNE1* locus, were modestly associated with better and worse prognosis, respectively, given the available sample size and adjustment for multiple testing (Supplementary Data 14). *WRN* deletion and *BRCA2* SNVs were also informative to the elastic net model, although the evidence for their association with overall survival is limited in these data (Supplementary Fig. 11). *CDK12* SNV, deleterious mt SNVs and severe chromothripsis were linked with overall survival in stage 3 tumours alone, which is consistent with effects that are independent of stage at diagnosis.

## Discussion

We have shown that the global landscape of structural variation in HGSOC is shaped by the presence of HRD and WGD, leading to the emergence of hotspots, impacting thousands of genes recurrently across samples. Previous studies have identified similar genomic regions but have not refined these regions to identify candidate driver genes[6,8,42]. Exploiting the independent sub-cohorts underlying the combined cohort, we predicted 8 HGSOC candidate driver genes (*PCM1, WRN, LEPROTL1, ARID1B, FGFR1OP, AXIN1, DNMT3A, TCEA1*) within these regions, showing significant differential expression across sub-cohorts in response to the CNA loads observed. Of these potential CNA-mediated driver events, only *WRN* (a DNA helicase involved in double-strand break repair) deletion showed some evidence of an effect on patient survival. Supporting this conclusion, *WRN* has recently been reported as a haploinsufficient tumour suppressor gene, based upon analyses of pan-cancer CNA data not examined here and experiments in lung epithelial cells[36]; we propose that it may also represent a therapeutic target in HGSOC. Other CNA-mediated driver gene candidates provide additional insights into HGSOC biology. Frequent CNA amplification of *DNMT3A* results in significantly higher expression, which has been reported in previous studies of HGSOC[43]. This suggests a role for *DNMT3A* in the aberrant DNA methylation

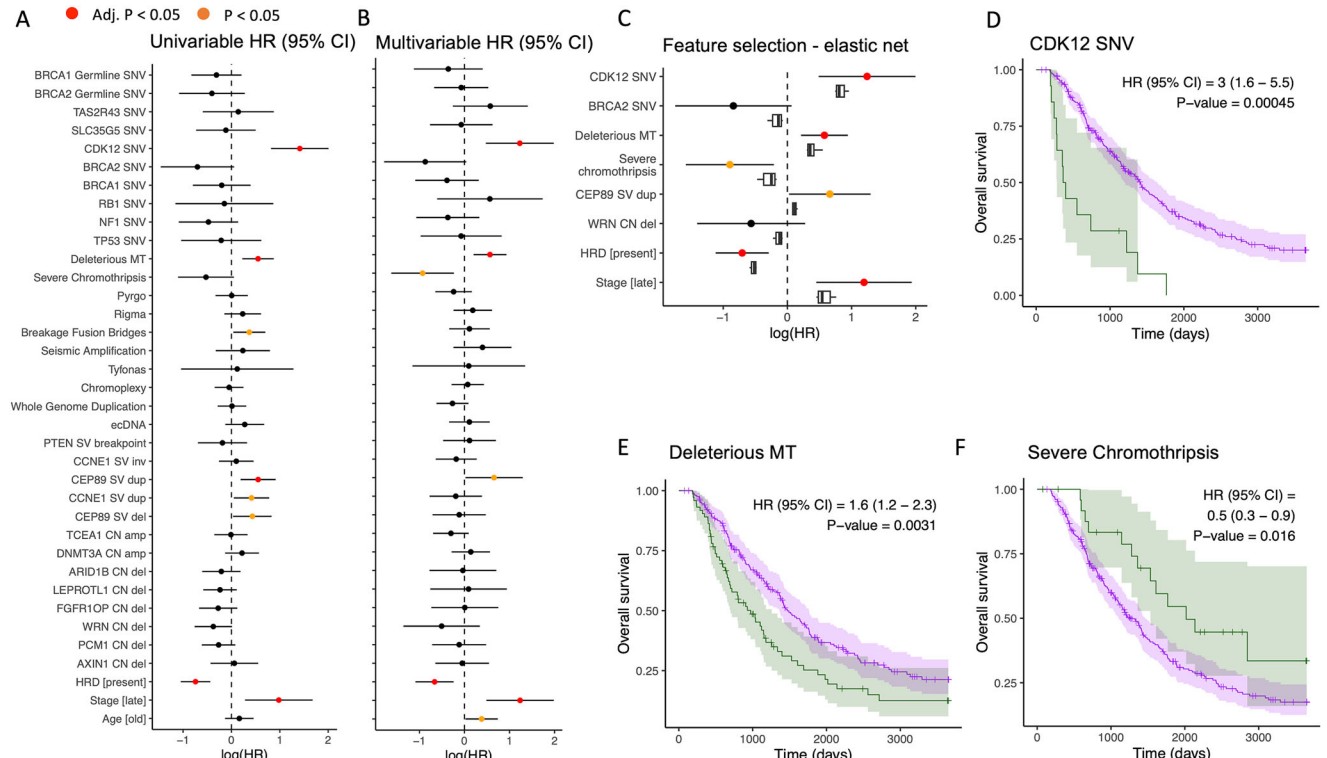

**Fig. 6 | Multivariable modelling of the impact of genomic features of HGSOC on overall survival adjusted for baseline clinical factors. A** Univariable modelling of 33 genomic features using a Cox Proportional Hazards (PH) model adjusted for HRD, age at diagnosis and stage at diagnosis and stratified by cohort (*N* = 277 samples with complete overall survival time and tumour stage data). Forest plot shows log hazard ratios (HRs) and 95% confidence interval (CI) per feature; *p*-values adjusted for multiple testing, the null hypothesis HR = 1. **B** As (**A**) but HRs from one multivariable model including all 33 genomic features plus adjustments. **C** HRs with 95% CI for selected features from elastic net penalised Cox PH regression model

(round points coloured as in B), versus boxplots for HR estimates (centre = median, box = 25% and 75% percentiles, whiskers = 1.5*IQR) from 10 cross-validations of elastic net for the same features. Kaplan–Meier (K–M) plots (*N* = 279 patients) show overall survival with 95% CI for presence (green curve) and absence (purple curve) of **D** CDK12 SNV, **E** deleterious mtDNA mutation **F** severe chromothripsis (>2 chromosomes affected by chromothripsis). Cox PH HRs and 95% CI are reported for each K–M plot, with p-value rejecting HR = 1 (*N* = 279 patients). Survival data are in Supplementary Data 12 and 14.

patterns seen in HGSOC, which are not currently well understood, but show promise as biomarkers for detection and prognostic testing[44]. To our knowledge, *ARID1B*, involved in chromatin remodelling, has not previously been reported to be frequently altered in HGSOC, although it has been reported to be inactivated in endometrial, endometrioid ovarian and clear cell ovarian cancer[45]. Reductions in the expression of Wnt/β-catenin inhibitors, such as *AXIN1*, could be a mechanism underlying previously reported pathway activation[46]. We have found recurrent CNA significantly altering *AXIN1* expression, but intriguingly, the rates of deletions and amplifications across samples are similar.

The extent of enrichment of SVs of all types, both simple and complex, on chromosome 19 is striking, constituting a chromosome-wide hotspot for complex structural variation. This enrichment is driven by patterns of fold-back inversions at the *CCNE1* locus[23,29], resulting in a higher level of amplification than possible by simple duplication and is achieved via complex mechanisms such as breakage-fusion-bridge cycles, ecDNA or chromothripsis. *CCNE1* amplification has been proposed as an effective therapeutic target[47]. However, the mechanisms leading to amplification and, in particular, overexpression of the gene are not fully understood. We show that SVs involving *CCNE1* are associated with its increased expression, and our data suggest that concurrent duplication of nearby gene *CEP89*, which likely reflects the same amplification events, may be linked with poorer prognosis. The detection of complex SVs based on informatics classifications from whole genome sequencing remains imperfect, and informing existing gold standards by incorporating more information on the mechanisms underlying these events is a necessary area of

future development if we are to accurately capture their prognostic value.

WGD is a common event in many tumour types, including HGSOC, and has been associated with poor prognosis across cancer types[11]. WGD promotes further chromosomal instability, increasing genetic diversity, tumour adaptability and the potential for cells to metastasise[48,49]. Our results confirm the association between WGD and genomic instability seen in cell line experiments[50], but extend this to encompass most known cSV types in addition to simple structural variation. It has been hypothesised that WGD may also allow rapid tumour evolution via catastrophic events such as chromothripsis[51], and we conclude there is convincing evidence for this in HGSOC. We have shown that those tumours undergoing WGD suffer frequent catastrophic events, particularly chromothripsis and BFB, and would be expected to evolve rapidly. This may represent an advantage for some WGD tumours, for example those acquiring BFB mediated amplifications of *CCNE1*, but appears to be a liability for those suffering the most severe chromothripsis events, which may contribute to increased immunogenicity or compromised metabolism, and are associated with longer overall patient survival (Fig. 6). A recent study found that HGSOC samples frequently showed evidence of chromothripsis but these events rarely caused losses of tumour suppressor or DNA repair genes[13]. Our results also suggest that these events do not generally fuel adaptive evolution and instead make up part of the deleterious mutation burden afflicting these tumours. This is consistent with simulations suggesting that WGD may be selected to mitigate the accumulation of deleterious alterations suffered by

tumours with high mutation rates[52]. In our HGSOC study, we see little evidence of a link between WGD specifically and overall patient survival. This may reflect the delicate balance of instability where WGD tumours can either benefit or be made vulnerable by the increased genomic scarring due to chromosomal instability. Nevertheless, the frequent occurrence of WGD has clinical significance, since WGD itself has been reported to be a targetable vulnerability[53,54]. This highlights the potential for therapeutic opportunities in patients with WGD tumours, which are generally HR proficient and currently more challenging to treat. A previous study by Lahtinen et al.[55], highlighted three mutation-based evolutionary trajectories in HGSOC, which were defined by clonal complexity and divergence. Two of these trajectories were enriched for WGD, but HRD status and copy number profiles were consistent between the groups. This suggests that our ability to stratify HGSOC may be enhanced by incorporating clonal dynamics into our study of complex genomic variation. Our knowledge of the cSV landscape is incomplete, and the study of these variants is rapidly developing. Despite our increased power to characterise the cSV landscape in this larger cohort, the current known cSV types encompassed only a small fraction (19%) of the total SVs observed in the cohort. A higher proportion of SVs (27%) than those included in cSVs are clustered in the genome[26], which can indicate cSV. This suggests that as yet unstudied cSVs may occur in HGSOC. Outside HGSOC, interrogation of the Hartwig pan-cancer cohort demonstrated at least one complex event in 95% of metastatic tumours[56], using similar methods to those implemented here. Large clusters of SVs were also present in other tumour types, similarly known to demonstrate HRD (breast and prostate). However, comparing the complex SV landscape between studies remains challenging due to the rapid evolution of methodologies to detect and interrogate these events[17].

We show that *CDK12* inactivation occurs at unexpectedly high levels in HGSOC, affecting up to 34% of samples in the cohort (Fig. 3C) when all structural variation is added to the deleterious SNV/SV/CNA load affecting 16% of samples (Fig. 3A). *CDK12* loss in HGSOC leads to genomic instability, in the form of extensive tandem duplication[57] and reduced expression of DNA damage repair genes including HR genes such as *BRCA1*[58]. The extent to which *CDK12* loss confers sensitivity to single-agent PARP inhibition remains contentious; in prostate cancer, the tandem duplication resulting from biallelic *CDK12* loss results in increased neoantigen generation and enhanced sensitivity to immunotherapy[59]. We show that *CDK12* mutation is associated with significantly worse overall survival (Fig. 6D), and it is possible that these patients, with genetic *CDK12* inhibition, may show major improvements in overall survival with PARPi treatment.

Finally, we demonstrate that deleterious SNV loads predicted to disrupt mitochondrial gene function accumulate in HGSOC and are a biomarker of poorer OS, independent of other influential variables such as HRD. Recent studies have revealed driver roles for mtDNA mutations during tumorigenesis in particular tumour types, but the functional impact of these variants on mitochondrial function is understudied[20]. Notably, deleterious mtDNA mutations in colorectal tumours are associated with improved OS[19], highlighting the tumour type-specific impacts of these mutations and the pressing need for further studies. There are two clinical implications of our observations in HGSOC. Firstly, disrupted mitochondrial function may be a targetable feature, particularly in WGD tumours, and therapeutic strategies to rescue mitochondrial CI deficiency are already under investigation in the context of cardiovascular disease[60]. Secondly, our data suggests that HRD tumours are intolerant of deleterious mtDNA mutations, consistent with their sensitivity to disrupted oxidative phosphorylation metabolism[61].

Overall, these data show that the genomic chaos seen in HGSOC obscures meaningful underlying patterns. Structural alterations are distributed non-randomly to generate hotspots harbouring known driver genes and other compelling candidates. The diverse and frequent complex structural events observed relate to the presence of two genomic states, HRD and WGD, which generate structural diversity but also create vulnerabilities for tumours. Epitomising this dichotomy, tumours with WGD are more likely to possess *CCNE1* amplifications enhancing proliferation, but are also more likely to suffer extreme chromothripsis, which appears to impair tumour development. Thus, WGD tumours walk a narrow path towards an optimal level of chromosomal instability, which facilitates rapid growth without risking cell death. These heavily disrupted nuclear genomes are, in turn, associated with alterations to the mitochondrial genome, impacting patient survival and revealing potential therapeutic targets.

## Methods

### Scottish sample collection and preparation for WGS and RNAseq

Scottish HGSOC samples (subsequently referred to as the SHGSOC cohort) were collected via local Bioresource facilities in Aberdeen, Dundee, Edinburgh and Glasgow as previously described, based upon written informed consent for the use of patient samples[24]. Clinical data for the SHGSOC cohort was retrieved from the Cancer Research UK Clinical Trials Unit Glasgow, the Edinburgh Ovarian Cancer Database and available electronic health records; the study received institutional review board approval from the Lothian Annotated Human BioResource (ethics reference 15/ES/0094-SR751) and NHS Greater Glasgow & Clyde Biorepository (ethics reference 22/WS/0020). HGSOC diagnosis was confirmed by formal expert pathology review (CSH), and samples were estimated to have >40% tumour cellularity by macroscopic visual assessment. Matched germline DNA was extracted from whole blood for each patient. Somatic and germline DNA were extracted from the tumour and blood, respectively, as described previously[24]. Somatic RNA was extracted from the same tumour sample as the DNA used for WGS. RNAseq was carried out by the Edinburgh Clinical Research Facility on an Illumina NExtSeq500 as previously described[24].

### Sequence acquisition and primary processing

WGS and RNA-seq reads were downloaded in compressed FASTQ format from the sequencing facility or in aligned BAM format (including unaligned reads) from the European Genome/Phenome Archive. The reads obtained in BAM format were query-sorted and converted to FASTQ. WGS reads were aligned to the GRCh38 reference genome. Sample quality control was performed with Qsignature 0.1 to identify sample mix-ups and VerifyBamId 1.1.3 to identify sample contamination.

RNA-seq data was analysed using the Illumina RNA-seq best practice template. Briefly, reads were aligned to the hg38 reference genome, and quality control was carried out. Salmon quant was used to quantify the expression of transcripts against the Ensembl 99 hg38 RefSeq transcript database, indexed using the salmon index (k-mers of length 31). Transcript-level abundance estimates were imported into R and summarised for further gene-level analyses. For differential expression analyses, raw expression counts were used by the DESeq2 package. Previously published RNA-seq data available for the AOCS[8] ($N = 80$), TCGA[6] ($N = 31$) and MDA[25] ($N = 26$) cohorts, together with RNA-seq data for the SHGSOC ($N = 69$) cohort, generated for the present study as detailed above, were processed in this way from FASTQ (overall $N = 206$).

### Single-nucleotide, indel, copy number and structural variant calling

Somatic and germline variant calling was performed using a bcbio[62] 1.0.7 pipeline as previously described[24]. Germline SNPs and indels (SNVs) were called with GATK 4.0.0.0 HaplotypeCaller. Germline SNVs reported in ClinVar[63] to disrupt the function of 12 HGSOC risk genes[64–66] were recorded (Supplementary Data 2), and *BRCA1/2*

variants were found to be enriched in patient samples relative to comparable populations in gnomAD[67]. Somatic SNVs and indels were called as a majority vote between Mutect2 1.1.5, Strelka2 and VarDict 2017.11.23. Small variants were annotated with Ensembl Variant Effect Predictor v91 and filtered for oxidation artefacts by GATK 4.0.0.0 FilterByOrientationBias. Somatic structural variants were called with Manta 1.2.1 and GRIDSS 2.7.3. Somatic copy number alterations (CNAs) were called with CNVkit 0.9.2a0, CLImAT 1.2.2 and PURPLE 2.51. In both cases, the intersection of the calls was taken forward as consensus calls. Consensus SVs were identified using viola-sv 1.0.0.dev10[68] with a proximity threshold of 100 bp. Where SVs are interrogated, this complete callset is used unless it is specified that SVs occurring in clusters (see methods lines 615–619) are analysed separately. Consensus CNAs had at least 50% overlap between segments with the same direction of copy number change. Tumour cellularity was estimated using both CLImAT's estimates and p53 variant allele frequency, and CLImAT's estimates were used to account for cellularity in differential expression analyses. To predict the level of HR deficiency in each tumour sample, we implemented the HRDetect algorithm as published by Davies and Glodzik et al.[69] as previously described[24]. Fusion genes were predicted in each sample by STAR-Fusion[70] based upon RNAseq data, then compared with the locations of Manta and GRIDSS SV breakpoints based upon WGS data. Although 1757 fusion transcripts were found, 90% were found in a single sample, and very few, seen more frequently, were within 1 kb of an SV breakpoint (Supplementary Data 15).

## Complex SV detection

Eight categories of cSV were predicted: chromothripsis, ecDNA, breakage fusion bridges, tyfonas, pyrgo, rigma, chromoplexy and seismic amplification. ShatterSeek[13] was used to call regions subject to chromothripsis[71] based on consensus SV and copy number calls, and following the recommended thresholds for the number of interleaved SVs, the number of adjacent segments oscillating between CN states, the number of interchromosomal translocations, the fragment joints test, the chromosomal enrichment test and the exponential distribution of breakpoints test[13]. Additionally, all chromothripsis calls were visually inspected, as is the current standard in the field, and to mitigate against false positives driven by the high baseline rates of instability in HGSOC. Candidate ecDNAs were predicted using AmpliconArchitect[72] with default settings to call ecDNA based upon purity and ploidy-adjusted somatic CN calls from PURPLE. The resulting graph and cycle output files were processed by AmpliconClassifier[73] to identify amplicons with CN > 4 and size >10 kb as ecDNAs. The junction-balanced genome graph (JaBbA) inference algorithm was used to generate genome graphs[15]. The resulting graphs were assessed as known cSV classes within gGnome[74], including breakage fusion bridges, tyfonas, pyrgo, rigma and chromoplexy. We detected putative regions of seismic amplification using an established approach[16] and the default threshold for amplification (CN ≥ 5 for diploid samples, and CN ≥ 9 for samples with ploidy >2). A candidate seismic amplicon was defined as one that contains amplified CN segments linked by ≥14 SV rearrangements, as recommended[16]. Visual examples of all classes of complex SVs are presented in Supplementary Fig. 4. To identify clusters of SVs, we used the same approach as used by the PCAWG consortium (clusterSV[26]). This approach identifies groups of SVs that occur closer together on a given chromosome than you would expect for that SV type on that chromosome, given the overall distribution of SVs.

## Identification of genomic regions of recurrent copy number alteration/structural variation

Regions of the genome undergoing recurrent copy number alteration were identified using GISTIC (v2.0.23). GISTIC compares the observed frequency of alteration in the region across all samples, combined with the observed magnitude of change, to the background expectations of

frequency and magnitude of copy number alteration obtained by permuting genomic regions. We considered regions to be significantly recurrently deleted or amplified at a $q$-value < 0.05, and the region boundaries were defined to ensure 95% confidence that the reported region contained the recurrent event of interest. The CNA hotspot on chromosome X was identified by separate GISTIC analysis. GISTIC was applied to PURPLE segmentation calls only, as these were more appropriately sized for use with GISTIC. More stringent filtering to consider only consensus calls within GISTIC peaks was applied downstream when investigating the impact of potential CNA drivers on expression.

Genomic regions of significant (FDR corrected $p$-value < 0.05) SV enrichment were identified using a negative binomial regression model of SV density throughout the genome, split into 50 kb bins with 1 kb overlap. All types of SV were considered separately in addition to breakpoint density. This model was implemented using the Fishhook package[32], adjusting for mappability (scores in 1 Mb bins), GC content, CpG islands[75], gene density (over 5 Mbp windows; GENCODE v41), the presence of repetitive elements (DNA transposons, SINE, LINE, short tandem repeats and long tandem repeats), and fragile sites[76]. Genomic regions with low mappability including callable sites with mappability score <0.9[77] and blacklisted regions[78] were excluded. Chromosomal enrichments of SV classes were assessed using binomial tests of proportions accounting for chromosome length.

## Prediction of SNV-mediated driver genes

We inferred SNV driver variants across the coding (canonical transcripts only) and non-coding genome. We used four driver prediction methods: OncodriveCLUSTL v1.1.3[79], OncodriveFML v2.2.0[80], ActiveDriverWGS v1.2.0[81] and dNdScv[82], with differing approaches to driver identification. Prior to the identification of driver variants, single-nucleotide variants were filtered to remove variants with variant allele frequency (VAF) < 0.1. OncodriveCLUSTL hyper-parameters (simulation window, smoothing window, and clustering window) were optimised by selecting from a range of values for each, to maximise: the goodness of fit of observed $p$-values to the null distribution and the enrichment of known cancer genes in candidate driver elements. Otherwise, we used default parameters. Candidate drivers were identified at thresholds of $Q$ < 0.01 for OncodriveFML and OncodriveCLUSTL, FDR < 0.05 for ActiveDriverWGS and gglobal cv < 0.01 for dNdScv <0.01. Overall, there was poor agreement between candidate drivers across algorithms (Supplementary Data 11). Six known HGSOC driver genes identified by dNdScv (expected to be the most conservative approach), plus two additional candidate genes identified by dNdScv and at least one other algorithm, were taken forward as candidate SNV-mediated driver genes, and the full list of candidates from all algorithms is provided.

## Prediction of CNA/SV-mediated driver genes

Genomic windows enriched for deletions, duplications, translocations, inversions and all breakpoints (at FDR < 0.05 by Fishhook) and CNA hotspots identified by GISTIC were intersected with COSMIC Cancer Gene Census (CGC) genes (version 96) and with the corresponding sample-level consensus SV and CNA calls, respectively. The CGC[27] is an established starting point for studies of cancer driver genes, containing 719 manually curated cancer driver genes, supported by functional validation studies and evidence of recurrent SNV/CNA in tumours. CGC genes in hotspots were tested for differential expression on the occurrence of the SV/CNA type driving the hotspot using DESeq2. Samples in which the SV/CNA intersects the CGC gene were compared with those lacking an SV/CNA of that type in the gene being tested. Differential expression was tested separately in each of the four cohorts for which RNA seq data were available, and in a combined, batch-corrected table of expression values combining cohorts. An adjusted $p$-value of 0.05 for the gene being tested in the DESeq2 results

was considered significant. The log2 fold change reported is that from the combined expression analysis, with the number of cohorts with a significant change reported as a replication score. An analysis of the FDR of this procedure suggested replication across cohorts maintained an FDR of less than 0.05.

Subsequently the GO/KEGG term enrichment (adjusted for multiple testing) among significantly differentially expressed genes observed between samples with and without a given driver variant was assessed with the R package clusterProfiler, using the methods enrichGO() and enrichKEGG(), based upon one sided Fisher's tests (for terms with greater than expected DE genes), *p* values were adjusted by the Benjamini Hochberg method, cutoff 0.05.

### Analyses of mutual exclusivity and co-occurrence patterns across somatic alterations

Complex SVs were clustered according to their prevalence across samples using hierarchical clustering based on Euclidean distances and using complete linkage. The major axes of variation in the prevalence matrix (excluding HRD and WGD) were identified using principal components analysis. Enrichment of complex SV types in HRD samples was calculated using logistic regression, and odds ratios were reported with 95% confidence intervals. Patterns of pairwise co-occurrence and mutual exclusivity between complex SVs were tested based on weighted mutual information using SELECT[83] to extract significant pairwise relationships relative to expectation, whilst limiting the false discovery rate. This was repeated to consider all pairwise combinations of key nuclear somatic alterations, HRD and mtDNA mutations by mitochondrial complex. The fraction of the tumour genome duplicated was compared in samples with and without breakage fusion bridges or chromothripsis, and the significance of the difference was tested using a Wilcox Rank-Sum test.

### Calling mtDNA mutations and mtDNA copy number

For each sample, reads aligned to the mitochondrial reference genome were extracted from the alignment files using Samtools (v1.12). The RtN! algorithm was used to mitigate against false positive variant calls from mismapped reads originating from nuclear-encoded mitochondrial pseudogenes (known as NUMTs)[84]. Variant calling was performed using VarScan2 (v2.4.4)[85], the preferred variant caller based on our own benchmarking analysis and its successful use in previous studies[18,86]. To estimate the number of copies of mtDNA in tumour and normal samples, we applied a formula derived from the pan-cancer analysis of the whole-genome study (PCAWG)[18]. Additionally, we implemented a stringent set of filters. These filters excluded variants displaying significant strand bias (phred > 60), variants located within error-prone regions due to homopolymers, variants with limited supporting reads (<30), and variants with extremely low heteroplasmy (<0.25% VAF). Subsequently, SNVs and INDELs were annotated using Ensembl's Variant Effect Predictor (VEP) version 107. mtDNA variants were also called in matched RNAseq samples to ensure the removal of NUMT-generated reads, since NUMTs lack evidence of transcription.

### Multivariable analyses of the impact of molecular factors on overall survival

The survival analyses considered 277 samples with complete overall survival time after diagnosis and tumour FIGO stage data. These samples were obtained from four cohorts (AOCS N = 80; BCCA N = 59; SHGSOC N = 110; TCGA N = 31). Overall survival times greater than 10 years were right censored (19 samples). The presence of: nine classes of complex SV (whole genome duplication, chromothripsis, pyrgo, chromoplexy, breakage fusion bridge, ecDNA, rigma, tyfonas and seismic amplification); SNVs in eight candidate driver genes (*TP53, NF1, RB1, BRCA1, BRCA2, CDK12, SLC35G5* and *TAS2R43*); CNA or SVs in eleven candidate driver genes (*AXIN1, PCM1, WRN, FGFR1OP, LEPROTL1, DNMT3A, ARID1B, TCEA1, PTEN, CCNE1* (inversions and

duplications), and *CEP89* (deletions and duplications)); and deleterious mitochondrial mutations were tested for their impact on survival. Overall survival was modelled using a Cox proportional hazards regression model using the coxph method in the R package survival. The effect of each variable was examined individually and in combination using a multivariable model adjusted for age (greater than mean age at diagnosis), FIGO stage (3 or above), HR deficiency as predicted by HRDetect, and stratified by cohort. Effect estimates with 95% confidence intervals were presented as forest plots. In addition, we applied cross-validated elastic-net regularised Cox regression to perform variable selection using the cv.glmnet method in the R package glmnet, revealing the most informative variables in the reduced forest plot.

### Statistics and reproducibility

No statistical method was used to predetermine sample size. No data were excluded from the analyses except where stated.

### Reporting summary

Further information on research design is available in the Nature Portfolio Reporting Summary linked to this article.

## Data availability

All WGS and RNA-seq data are available from the European Genome/Phenome Archive (EGA) and the US National Cancer Institute (NCI) as follows. Australian Ovarian Cancer Study (AOCS)[8]: EGAD00001000293 [https://www.ega-archive.org/studies/EGAS00001000397], British Colombia Cancer Agency (BCCA)[23]: EGAD00001003268, MD Anderson (MDA)[25]: EGAD00001005240 [https://ega-archive.org/dacs/EGAC00001001288] are available under restricted access due to data privacy laws; access can be requested from their respective Data Access Committees via EGA. The Cancer Genome Atlas (TCGA)[6]: NCI Genomic Data Commons [https://portal.gdc.cancer.gov/]. The clinical information for the AOCS and TCGA patients is available as part of the PCAWG project[41]. Clinical information for the MDA and BCCA cohorts is available from the supplementary data of their respective publications[23,25]. All SHGSOC cohort data (WGS and RNA-seq) generated for this study are available at accession number EGAS00001004410 under restricted access due to (UK GDPR) data privacy laws. Access can be arranged for researchers for 12 months by a simple request to the Data Access Committee (DAC) via EGA; the DAC will reply to requests within one week. Source data are provided with this paper.

## Code availability

Computational and statistical code, including all code used to generate the figures, is available at the following GitHub repository: https://github.com/EwingGroup/TrajectoriesHGSOC. Primary processing code is available here: https://github.com/ailithewing/Structural_variants_BRCA1_2_HRD_inHGSOC.

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

## Acknowledgements

AE is supported by a University of Edinburgh Chancellor's Fellowship, a Langmuir Talent Development Fellowship and core funding from the MRC Human Genetics Unit. C.A.S. and A.M. are supported by MRC core funding to the MRC Human Genetics Unit, University of Edinburgh and MRC Programme funding MC_UU_00035/1. E.E.B. is an MB-PhD student supported by funding from CRUK (TRACC Programme SEBCATP-2022/100007). A.H. is supported by the Edinburgh Clinical Academic Training programme (CRUK grant C157/A29279). J.T. received core centre funding from CRUK. P.R. received research funding from the Beatson Cancer Charity. Sequencing of the SHGSOC cohort was supported by AstraZeneca, the Medical Research Council, and the Scottish Chief Scientist through a Precision Medicine Scotland Innovation Centre/Scottish Genome Partnership (SEHHD-CSO1175759/2158447) collaboration. This Scottish Genomes Partnership was funded by the Chief Scientist Office of the Scottish Government Health Directorates (SGP/1) and The Medical Research Council Whole Genome Sequencing for Health and Wealth Initiative (MC/PC/15080). This study would not be possible without the families, patients, clinicians, nurses, research scientists, laboratory staff, informaticians and the wider Scottish Genomes Partnership team to whom we give grateful thanks. Members of the Scottish Genome Partnership (SGP) include Timothy J. Aitman, Andrew V. Biankin, Susanna L. Cooke, Wendy Inglis Humphrey, Sancha Martin, Lynne Mennie, Alison Meynert, Zosia Miedzybrodzka, Fiona Murphy, Craig Nourse, Javier Santoyo-Lopez, Colin A. Semple, and Nicola Williams. More information about SGP can be found at www.scottishgenomespartnership.org. The authors would also like to acknowledge the Edinburgh Clinical Research Facility for the sequencing of RNA samples from the SHGSOC cohort. The authors would also like to extend their thanks to the Nicola Murray Foundation and the Edinburgh Ovarian Cancer Database, from which the clinical data for much of the Scottish cohort were retrieved. We also thank the NRS Lothian Human Annotated Bioresource, NHS Lothian Department of Pathology, the Edinburgh Experimental Cancer Medicine Centre, the Biorepository at the Glasgow Queen Elizabeth University Hospital, and the Tayside Biorepository for their support. This manuscript was prepared using a limited-access dataset obtained from BC CANCER and does not necessarily reflect the opinions or views of BC CANCER. For the purpose of open access, the author has applied a Creative

## Author contributions

A.E., C.A.S. and C.G. conceived and designed the analysis of this cohort. C.G., C.A.S., C.S.H., I.A.M., T.M, M.F., N.M, A.M, B.D., R.M., J.C.B., A.V.B. and L.M. conceived the study within the broader remit of the Scottish molecular ovarian cancer collaboration. M.C., R.L.H., M.F., M.L., T.M., N.M., N.S., M.M., F.N., R.G., P.R. and C.G. acquired patient samples. C.S.H. and R.L.H. performed histopathological review. M.C., I.C., D.E. and B.M. performed sample processing. B.M. and A.V.B. performed whole genome sequencing. A.Me, A.E., G.R.G., S.A., R.S., S.B. and T.B. processed the sequencing data. R.L.H., C.B., T.M., M.F., N.M., M.L., R.G., M.M., F.N., P.R. and C.G. provided clinical data. A.E., R.S., S.A., D.P.B., S.L.B., A.H., J.M., J.P.T., K.S., T.P. and E.E-B performed downstream statistical and computational analyses. A.E., P.R., C.G. and C.A.S. provided strategic direction to the project. SGP and AstraZeneca funded the work. A.E., C.A.S., P.R. and C.G. drafted the paper. All authors read and commented on the manuscript and approved the final version.

## Competing interests

C.G. receives research funding from AstraZeneca, MSD, Novartis, GSK, BerGen Bio, Medannex, Roche, Verastem, Artios and personal fees from AstraZeneca, MSD, GSK, Clovis, Verastem, Takeda, Eisai, Cor2Ed, Peer Voice. PR received honoraria from AstraZeneca. RH received consultancy fees from GSK and DeciBio. BD and RM are employees and stockholders of AstraZeneca. IMcN is or was previously on the advisory boards for Clovis Oncology, Tesaro, AstraZeneca, Carrick Therapeutics, Roche and ScanCell. IMcN also benefits from institutional funding from AstraZeneca. R.G. is or has been on the advisory boards of AstraZeneca, GSK, Tesaro and Clovis; has received speaker fees and funding to attend medical conferences from GSK and Tesaro and is a UK co-ordinating investigator or site principal investigator for studies sponsored by AstraZeneca, GSK, Pfizer and Clovis. The remaining authors declare no competing interests.

## Additional information

[1]MRC Human Genetics Unit, Institute of Genetics and Cancer, University of Edinburgh, Edinburgh, UK. [2]Cancer Research UK Scotland Centre, Institute of Genetics and Cancer, University of Edinburgh, Edinburgh, UK. [3]Nicola Murray Centre for Ovarian Cancer Research, Cancer Research UK Scotland Centre, Institute of Genetics and Cancer, University of Edinburgh, Edinburgh, UK. [4]Edinburgh Cancer Centre, Western General Hospital, NHS Lothian, Edinburgh, UK. [5]Edinburgh Pathology, Cancer Research UK Scotland Centre, Institute of Genetics and Cancer, University of Edinburgh, Edinburgh, UK. [6]Department of Oncology, University of Oxford, Old Road Campus Research Building, Roosevelt Drive, Oxford OX3 7DQ, UK. [7]Department of Oncology, Ninewells Hospital, NHS Tayside, Dundee, UK. [8]Division of Molecular and Clinical Medicine, School of Medicine, University of Dundee, Dundee, UK. [9]Tayside Biorepository, School of Medicine, University of Dundee, Dundee, UK. [10]Department of Oncology, Aberdeen Royal Infirmary, Aberdeen, UK. [11]Department of Oncology, Raigmore Hospital, NHS Highland, Inverness, UK. [12]Department of Gynaecological Oncology, Glasgow Royal Infirmary, Glasgow, UK. [13]Beatson West of Scotland Cancer Centre, Glasgow, UK. [14]School of Cancer Sciences, Wolfson Wohl Cancer Research Centre, University of Glasgow, Glasgow, UK. [15]Ovarian Cancer Action Research Centre, Department of Surgery and Cancer, Imperial College, London, UK. [16]Precision Medicine Scotland (PMS-IC), Queen Elizabeth University Hospital, Glasgow, UK. [17]Oncology Translation and Big Data Mining, GSK, Bishop's Stortford, UK. [18]Translational Medicine, Oncology R&D, AstraZeneca, Waltham, MA, USA. [19]Precision Medicine, Oncology R&D, AstraZeneca, Cambridge, UK. [20]West of Scotland Pancreatic Unit, Glasgow Royal Infirmary, Glasgow G31 2ER, UK. [21]South Western Sydney Clinical School, Faculty of Medicine, University of NSW, Liverpool, NSW 2170, Australia. [22]These authors contributed equally: Patricia Roxburgh, Charlie Gourley, Colin A. Semple. ✉e-mail: ailith.ewing@ed.ac.uk

## The Scottish Genomes Partnership

Andrew V. Biankin [14,20,21], Alison Meynert [1] & Colin A. Semple [1,22]

A full list of members and their affiliations appears in the Supplementary Information.

