## [Transparent Peer Review file · Nature Communications]

Divergent trajectories to structural diversity impact patient survival in high grade serous ovarian cancer

Corresponding Author: Professor Colin SEMPLE

Version 0:

Reviewer comments:

Reviewer #1

(Remarks to the Author)

In this manuscript, Ewing and colleagues performed genomic analysis on a cohort of over 324 high-grade serous ovarian cancer (HGSOC) genomes. The main focus is structural variation/genomic rearrangements that are a hallmark of HGSOCs. The WGS cohort was assembled from several previously published studies and the genomic data were processed by a harmonized workflow consisting of published methods. I believe the data are valuable and of broad interest to cancer genomic researchers. However, I have found issues related to (1) computational analysis; (2) presentation of the analytical results; and (3) interpretation of the genomic findings. Because of these issues, I do not recommend publication of the manuscript at Nature Communications until some or all of these issues are adequately addressed.

Technical issues:

1. Complex SV detection. Given the prevalence of complex rearrangements in ovarian cancers, I consider this issue to be important for the validity of many conclusions.

1.a. I have not found a description of how DNA rearrangement junctions are detected.

1.b. The authors commented that "all chromothripsis calls were visually inspected as is the current standard in the field." The authors should at least present some examples to justify the consistency of manual review.

1.c. In contrast to chromothripsis, the authors appeared to have relied on informatic tools (JaBbA, AmpliconArchitect, gGnome) for the classification of other types of complex rearrangements, without manual review. The authors should present some examples for each pattern and/or comment on the specificity of informatic

classifications.

1.d. It is not uncommon that the rearrangements on a single chromosome may show features of multiple classes of "cSVs". For example, some chromothripsis events can have segmental amplifications that may qualify as ecDNA amplification. The authors should discuss how such instances are handled.

1.e. The authors should describe the general criteria for classifying simple rearrangements (e.g., deletion, duplication etc.) and

translocations and how they are distinguished from complex rearrangements (e.g., Figure 2C).

2. Quantification of the burden of DNA rearrangements and copy-number alterations. The authors need to apply consistent criteria for the classification and quantification of copy-number alterations or rearrangements. For example, in Supp

Figure 1A, the median number of duplications is between 10 and 100, but in Supp Figure 1C, the median number of duplications is ~1000. How should one interpret these results? Moreover, different classes of alterations should be quantified separately. For example, gain or loss of a chromosome can arise from a single chromosome missegregation event; the same is true for chromothripsis. In Suppl Figure 2A, many samples have >1,000 "mutational events." Are the authors counting individual segmental deletions of a chromothripsis pattern as separate events, or grouping them as one instance of chromothripsis?

3. Enrichment analysis. It is unclear how the authors assess the frequency of a gene or locus being affected by rearrangement. For example, in Figure 2C, the distributions appear to be near identical for many types of rearrangements. Do these distributions simply reflect the distribution of breakpoints that are classified as one or multiple "cSV" clusters? For the enrichment of copy-number alterations, the authors did not distinguish between genes that may be disrupted as passengers. For example, in Figure 3A, disruption of CEP89 (3Mb from CCNE1) appears to follow the same pattern as CCNE1; LEPROTL1 (1Mb from WRN) shows a similar concurrence with WRN. I tend to think that their co-occurrence is attributed to the positive selection for the disruption of CCNE1 and/or WRN, but not due to LEPROTL1 or CEP89 being new drivers.

Presentation issues:

1. Figure 1. (1) Most rearrangements only affect genes that are adjacent to the breakpoints. The crude quantification based on "number of bases disrupted" is misleading. (2) Segmental copy-number alterations are accompanied by DNA rearrangements. How did the authors address such redundancy. (3) It would be much more informative to group patients by HRD status/Germline HRD mutations, Stage, and/WGD, and plot the burden of SNVs, SCNAs, and rearrangements that are sorted within each subgroup. (4) The authors need to describe how they identify genes being disrupted by mutations in Fig. 1B and

1C. The suggestion that 84% of all protein coding genes are disrupted by one of four types of alterations is unrealistic.

2. Figure 2. In Figure 2B, what is being used for clustering? Is it by genes, or regions?

3. Figure 3. Isn't all the information in C already shown in A?

4. Figure 4. A and B. How is the clustering performed? How are the PCs of complex SV signatures determined? C and D. What's new in D that is not contained in C?

Interpretation issues/further analysis needed:

5. CCNE1 is often amplified in HGSOV and tumors with CCNE1 overexpression show a high prevalence of large tandem duplications (Cancer Cell by Menghi et al). The authors should comment on if the same pattern is observed here.

6. How often do complex SVs lead to gene fusions? How does complex SVs or simple SVs alter gene expression after correcting for copy-number changes?

7. Do HRD or WGD ovarian cancers follow different evolutionary trajectories? Is there any difference between tumors from patients with germline HRD and those with HRD due to somatic mutations?

8. As the authors want to highlight the differences between WGD and HRD ovarian cancers. It would be desirable to present and contrast genomic patterns for WGD and HRD samples separately throughout the study.

9. It would also be broad interest if the authors can compare their findings in ovarian cancers to complex SVs (cSVs) in other types of cancers that show WGD or HRD (e.g., breast cancers).

Minor issues:

1. The main tumor genomic features, including CNV load and SV load, are confounded by tumor fraction. Please provide data to rule out this possibility.

2. Please provide details on how the copy number baseline was determined for gains or losses in samples with and without WGD.

3. Please ensure that the genomic features of each gene are corrected for clonality and that driver genes are involved in clonal events.

Most of the questions have not been adequately addressed with supporting evidence. While I believe the data are valuable and hold broad interest for cancer genomics researchers, the analysis and presentation remain misleading for the average reader. Due to these concerns, I do not recommend the manuscript for publication in Nature Communications.

Technical issues:

1. Complex SV detection. Given the prevalence of complex rearrangements in ovarian cancers, I consider this issue to be important for the validity of many conclusions.
 - 1.d. Some overlapping complex SVs are interrelated, and reporting both events separately could be problematic without clear supporting evidence.
2. Quantification of the burden of DNA rearrangements and copy-number alterations. Given the complexity of rearrangements, presenting data by simply combining SV burden and CNV burden can be misleading.
3. Enrichment analysis. Interpreting the conflicting results showing that hotspot regions around CCNE1 on Chr19 are enriched for both deletions and duplications is challenging. One possibility is that, in each sample, they arise from a single complex event that were misclassified.

Presentation issues:

1. Figure 1. Since most of the genes in Figure 2D fall within focal copy number alterations, we are not convinced that the expression of all genes spanning broad SVs (10 Mb or more) would be affected. This may have led the authors to overestimate that 84% of all protein-coding genes are disrupted in this cohort. As a result, Figure 1 presents a potentially misleading genomic landscape for average readers.

Interpretation issues/further analysis needed:

2. CCNE1 is often amplified in HGSOV and tumors with CCNE1 overexpression show a high prevalence of large tandem duplications
(Cancer Cell by Menghi et al). The authors have thoroughly addressed this question.
3. How often do complex SVs directly lead to gene fusions? Many tools have been developed to investigate the interactions between gene expression and copy-number changes, even at the single-cell level. Performing a correlation analysis would be beneficial.
4. Do HRD or WGD ovarian cancers follow different intra-patient evolutionary trajectories? If possible, including germline alterations related to HRD in Figure 1A would be helpful.
5. The authors have not clearly addressed the genomic differences between WGD and HRD ovarian cancers. Although WGD and HRD are not mutually exclusive, it would be helpful to illustrate the distinct genomic landscapes of these two groups in Figure 1A.
6. It would be of broad interest if the authors could present data in terms of comparing their findings in ovarian cancer to complex SVs observed in other cancer types with WGD or HRD.

Minor issues:

1. There is still a lack of evidence to demonstrate that key tumor genomic features, such as CNV load and SV load, are not confounded by tumor fraction. A correlation analysis would be helpful.
2. There is still a lack of detailed evidence on how copy number alterations are determined based on the copy number baseline in samples with or without WGD.
3. Determining the clonality of somatic mutations is not as challenging as the authors suggest. Many tools have been developed for this purpose.

Reviewer #2

(Remarks to the Author)

(Remarks on code availability)

I couldn't find any code for variant calling, which is essential for validating many of the conclusions.

Reviewer #3

(Remarks to the Author)

The authors state their impact is in improved understanding of driving events in high-grade serous ovarian cancer: "The driver landscape in HGSOC remains controversial, highlighting the need for comprehensive analyses of the mutational complexity of HGSOC genomes and elucidation of its clinical impact, in larger WGS cohorts"

NOTEWORTHY RESULTS:

**H RD and WGD underlie different evolutionary trajectories to structural diversity

**Deleterious SNV loads in mtDNA are a novel biomarker of poor patient prognosis

**SNVs in mitochondrial genes is higher in tumors with WBG and lower in tumors with HRD - implies less tolerance to mitoDNA mutations for HRD deficient tumors.

SIGNIFICANCE:

This work extends the established literature in outlining the molecular events that characterize high-grade serous carcinoma. They point out molecular factors that point to a poorer prognosis.

Are there any flaws in the data analysis, interpretation and conclusions? Do these prohibit publication or require revision?
Some imprecisions as noted below.

Is the methodology sound? Does the work meet the expected standards in your field? YES

Is there enough detail provided in the methods for the work to be reproduced? YES

LINE 88: most studies would support that the platinum sensitivity of ovarian cancer is quite a bit higher than "62%". You use as your reference ICON8 which was a study of patients randomized to neoadjuvant chemotherapy - these studies tend to enroll patients with a poorer prognosis and likely this impacts the platinum sensitivity response. Anyone who treats ovarian cancer will trip over the "62%" number and think it inaccurate. I would suggest saying something less definitive like "sensitive to platinum chemotherapy in 70-80% of cases", and quote GOG 111 (McGuire et al, NEJM, 70% RR), GOG 158 (Ozols et al JCO, 88%).

PAGE 12 / Figure 4 and associated text. The authors correctly note on lines 169-170 "HRD and WGD are not mutually exclusive but are anti-correlated such that HRD is depleted in samples with WGD (odds ratio (OR) 0.56; Chi-squared $p=0.015$)." However, on page 12 / figure 4 and associated text, they use the terms "strong mutual exclusivity" and "weak mutual exclusivity". This terminology seems imprecise given that mutual exclusivity is a binary condition. I would suggest they standardize their language here and categorize HRD and WGD (and other genomic events that they compare) as "strongly" or "weakly" ANTI-CORRELATED.

LINES 374-396: an assessment of factors impacting OS that concludes that FIGO stage drives prognosis is quite anticlimactic. This study is primarily made up of stage 3/4 tumors. I would be much more interested in a prognosis assessment that REMOVED FIGO stage as a factor (i.e. only assess stage 3C cancers - you have a huge number of them and could make potentially much more interesting associations by eliminating stage as a factor). BRCA germline mutation carriers are known to have a better prognosis. This data is available based on your figure 1. Its not clear to me in figure 6 why germline BRCA was not included in the univariate / multivariate analysis. If you excluded this because of the HRD assessment then why also include the BRCA SNV status in the analysis?

(Remarks on code availability)
out of time to do this. Quick review of the github file looks like all is in order.

Reviewer #4

(Remarks to the Author)

The authors describe the structural variations and driver genomic events in high-grade serous ovarian cancer. The manuscript is a tour de force for characterizing the structural variants in HGSC and is worth publishing after a few clarifications.

The study is largely descriptive, and most of the impact and scientific novelty lies in their large, unified dataset and descriptions of the mitochondrial mutations. However the methods are rigorous and the manuscript is well-written, figures are clear and comprehensive. Many of the findings are well aligned with previous work. E.g. Previous work has presented a larger ($n=510$) multi-site WGS dataset and found three divergent mutation-based evolutionary trajectories contributing to survival in HGSC (PMID 37207655). It would be interesting to discuss how these trajectories are distinct from the two trajectories described here.

WGD is also a known poor prognostic indicator and has been validated across several studies. This should be mentioned, as well as the papers referred to in the discussion.

The authors combine published WGS and gene expression data to construct a large WGS HGSOC cohort with matched RNA-seq to date (WGS $N=324$). I would refrain from superlatives in a scientific paper.

Supplementary Figure 1 has an overlapping graph - and is not evaluable.

(Remarks on code availability)

Version 1:

Reviewer comments:

Reviewer #1

(Remarks to the Author)

The authors have adequately addressed my questions regarding the technical aspects of their bioinformatic analysis. However, the authors have not been able to derive much novel insight from the results produced by the bioinformatics analyses. I therefore find the revised manuscript not reaching the level of publication at Nature Communications. My comments below center on the results presented in the main figures.

1. Figure 1A. The authors remain adamant about their way of quantifying the "number of bases disrupted," which I consider to be totally misleading. In Fig. 1A, top, there are several samples with $>6,000,000,000$ bases disrupted. Given that a diploid human cell contains 6 billion nucleotides from both set of chromosomes, this implies that more than every single nucleotide is disrupted, which is clearly nonsense. This form of presentation reflects a fundamental misunderstanding of genetic alterations. As the authors chose to sort the samples by their arbitrarily defined parameter of "number of bases disrupted," there is no clear pattern between the different types of complex DNA rearrangements, WGD, or HRD status.

2. Figure 1B. It remains unclear how the authors distinguish between genes disrupted by CNA and genes disrupted by structural variants. Segmental copy-number alterations (deletions or duplications) are invariably associated with DNA rearrangements. Are these counted separately, or added together? Judging from the right panel in Fig. 1B, the last row ("any") appeared to be close to the sum of the four numbers above. I consider this highly unlikely, since many rearrangements in ovarian cancers are associated with segmental copy-number alterations (therefore, the genes disrupted by such rearrangements should be classified under the CNA categories.)
3. Figure 1C. What insight has been provided by the list of most frequently disrupted genes except for TP53? Most of the genes are not known to contribute to carcinogenesis (including in ovarian cancers). I consider their frequent disruption to reflect their susceptibility to genomic/chromosomal instability. But that result does not need to be highlighted in a main figure panel.
4. Figure 2B. This is in my view one of a few informative figure panels in this study. What are the duplication and deletion hotspots being used for clustering? And What's the rationale for performing a clustering based on such hotspots? Judging from the result, I see that HRD samples show more deletions and WGD samples show more duplications. This pattern may be even more clear if the authors simply group samples based on HRD/WGD status, e.g., HRD (no WGD), HRD & WGD, and WGD (no HRD), and then sort by CNA burdens per sample.
5. Figure 2C. Is CCNE1 a deletion hotspot, duplication hotspot, or a SV hotspot? Are the authors referring to "deletion-type" or "duplication-type" SVs when they use "Deletion" or "Duplication"? There has been much confusion about these terminologies in the literature. I suggest the authors stay away from "deletion SV" or a "deletion type SV" but focus on relative copy-number status (i.e., gene copy number relative to the genome-wide average), and rearrangement breakpoints (regardless of their type or orientation).
6. Figure 2D. This figure panel is confusing for a variety of reasons. First, the use of SV deletion and CNA deletion etc. is confusing. Second, the meaning of the color of the labelled genes is not explained. Why are there two CCNE1 labels? Finally, the authors should annotate which genes have been implicated in ovarian cancer oncogenesis and which genes are likely affected because of their proximity to cancer genes.
7. Figure 3. This figure is essentially a re-cap of Figure 1. Despite the authors acknowledging that many of the genes listed in panel A are likely passengers, they still label them as "candidate driver genes." I consider this to be very misleading.
8. Figure 4. This figure can be potentially interesting. Figure 4A is essentially a repeat of the content in Fig. 1A. The meaning of the components in Fig. 4B is unclear to me. Fig. 4C and Fig. 4D are interesting and will benefit from being presented together. In Fig. 4E, it will be more clear to simply compare genomes with or without WGD, instead of relying on the % of genome duplicated as an indirect measure.
9. Figure 5. I cannot comment on this as I do not know about mitochondrial genetics.
10. Figure 6. I consider this figure to be the most novel result of the paper. It will be further improved if the authors eliminate passenger genes (i.e., those in close proximity to a known cancer driver) from the regression analysis. Panel D is a very clean result (although I'm unsure if similar results were reported previously). In Panel F, can the authors clarify their definition of "severe chromothripsis"? Is it based on the # of chromosomes inferred to have undergone chromothripsis, or the total number of breakpoints in chromothriptic rearrangements?

(Remarks on code availability)

Reviewer #2

(Remarks to the Author)

(Remarks on code availability)

Reviewer #3

(Remarks to the Author)

I have reviewed the response to reviewers and my concerns were appropriately addressed. I have no objection to the revised publication.

(Remarks on code availability)

Reviewer #4

(Remarks to the Author)

All my comments have been addressed. I support the acceptance of the paper for publication.

(Remarks on code availability)

Version 2:

Reviewer comments:

Reviewer #1

(Remarks to the Author)

No more comments to the authors. I respectfully disagree with the authors' presentation of their analytical results and do not think further exchange is worthwhile at this moment. I recommend solicitation of additional referees to evaluate my comments and the authors' revision and responses.

(Remarks on code availability)

Reviewer #2

(Remarks to the Author)

(Remarks on code availability)

Reviewer #5

(Remarks to the Author)

In this manuscript, Ewing et al. analyzed WGS and RNAseq data from 324 HGSO tumors (taken from different cohorts but analyzed uniformly). They describe the mutational landscape of HGSO, define candidate driver genes, determine the SV profiles, and also study mitochondrial mutations in these tumors.

The paper has already been reviewed by others, so I will mostly limit my review to my overall impression and to specific comments on Reviewer #1's concerns and the way in which the authors addressed them.

The manuscript is largely descriptive, and indeed doesn't shed much new light on the biology of HGSO. An interesting claim is that HRD and WGD underlie different evolutionary trajectories to structural diversity, however this claim is not fully substantiated – WGD and HRD are far from being mutually-exclusive, so it's difficult to classify tumors in this way. Another interesting finding is the prevalence of mtDNA alterations, but the biological meaning of this findings remains somewhat under-explored (for example, would samples with jeopardized mitochondria experience metabolic stresses induced by mitochondria dysfunction? The authors have RNAseq data from the samples but they haven't explored this at all). The most interesting findings are likely the strong enrichment for SVs on Chr19, and the association of chromothripsis with better prognosis in HGSO. Overall, I agree with Reviewer #1 that the biological insights from this work are quite limited.

Having said that, I find that the authors have reasonably addressed most of the specific concerns of Reviewer #1:

- (1) I agree with Reviewer #1 that the presentation of the alterations as the total number of bases affected doesn't have much sense – the authors have replaced this presentation in Fig. 1A as requested.
- (2) The authors explain what they've done in Fig. 1B, I think it's acceptable.
- (3) The authors address the Reviewer's point regarding the recurrently mutated genes.
- (4) The authors performed the clustering requested by the Reviewer.
- (5) I find the authors' response to this point acceptable.
- (6) I think this response is reasonable as well.
- (7) The Reviewer is right that this can be somewhat misleading, but the authors' rationale makes sense, and I wouldn't insist on their changing this Figure.
- (8) The new Fig. 4B is clear and this point has improved.
- (9) This is highly descriptive, but the inclusion of mtDNA mutations and the (albeit unsurprising) finding that they are so prevalent, are still novel and worth a Main Fig.
- (10) This is quite interesting, and I find the authors' response quite reasonable.

One major limitation of the study, which I don't know if came up during earlier rounds of review, is that the study under-utilizes the RNAseq data. The authors only use these data for the prioritization of candidate drivers, but it can be used in multiple additional and interesting ways. Two prominent examples:

- For novel candidate driver genes, is there any evidence that specific pathways in which they participate are dysregulated

in the tumors that carry these genetic alterations?

- As mentioned above, it could be telling to examine mitochondria-related gene expression in the context of the mtDNA mutations.

I'd encourage the authors to better take advantage of their matched RNAseq data, as this can potentially result in stronger biological insights and increase the enthusiasm toward the study.

(Remarks on code availability)

entitled “Divergent trajectories to structural diversity impact patient survival in high grade serous ovarian cancer” to Nature Communications. Thank you for taking the time to review our manuscript. We have endeavoured to address all comments in full as detailed below.

Reviewers 1 and 2

Technical issues:

3. **Complex SV detection.** Given the prevalence of complex rearrangements in ovarian cancers, I consider this issue to be important for the validity of many conclusions.

1.a. I have not found a description of how DNA rearrangement junctions are detected.

We have separated the variant calling methods into their own paragraph in the methods with subheading ‘Single nucleotide, indel, copy number and structural variant calling’ for ease of reference. Lines 578-582 detail the approach taken to SV calling. Somatic structural variants were called with Manta 1.2.1 and GRIDSS 2.7.3 and the intersect of the calls were taken forward as consensus calls. Consensus SVs were identified using *viola-sv* 1.0.0.dev10 with a proximity threshold of 100bp.

1.b. and 1.c. The authors commented that “all chromothripsis calls were visually inspected as is the current standard in the field.” The authors should at least present some examples to justify the consistency of manual review.

In contrast to chromothripsis, the authors appeared to have relied on informatic tools (JaBba, AmpliconArchitect, gGnome) for the classification of other types of complex rearrangements, without manual review. The authors should present some examples for each pattern and/or comment on the specificity of informatic classifications.

We have included visual examples of all types of complex SV considered in this study as an additional Supplementary Figure 4. Robustly identifying complex SVs informatically from whole genome sequencing data remains a challenge for the field. In all cases we have applied the current gold standard approaches to identifying these events at scale. We have included a sentence in the discussion highlighting that existing methods are imperfect and a necessary area of future development in the field (Lines 449-452).

In the case of chromothripsis, the authors of ShatterSeek recommend an additional manual review step to ensure the hallmarks of chromothripsis are present (e.g. copy number oscillation, breakpoint enrichment, random distribution of SV types). Therefore, we reviewed ~500 chromothripsis calls to ensure the hallmarks could be easily visualised. However, for other complex SV types where the hallmarks are more ambiguous and less easy to visualise we relied on established informatic methods alone.

Institute of Genetics and Cancer

The University of Edinburgh, Western General Hospital, Crewe Road, Edinburgh EH4 2XU

T: +44 (0)131 651 8500 ed.ac.uk/institute-genetics-cancer @EdinUni_IGC

1.d. It is not uncommon that the rearrangements on a single chromosome may show features of multiple classes of “cSVs”. For example, some chromothripsis events can have segmental amplifications that may qualify as ecDNA amplification. The authors should discuss how such instances are handled.

It is true that 298/776 (38%) of regions with predicted complex SVs overlap with at least one other complex SV region in our data. To reflect the challenge in discriminating between possible underlying mechanisms in the genomic signal of such events, we have included the regions with multiple overlapping calls as both classes of complex SV in the analyses. We have clarified this in lines 198-204 of the revised manuscript.

When we re-analyse the data to exclude complex SV calls where we are unable to discriminate between possible underlying mechanisms, we still observe broadly consistent positive correlations between complex SVs and WGD and negative correlations with HRD (e.g. Figure 4C). This is despite the substantial reduction in statistical power from excluding these events.

Alternative Figure 4C: This is figure 4C replotted excluding complex SVs whose genomic footprint overlaps with another complex SV call of a different type.

1.e. The authors should describe the general criteria for classifying simple rearrangements (e.g., deletion, duplication etc.) and translocations and how they are distinguished from complex rearrangements (e.g., Figure 2C).

Throughout this study, SVs (e.g. SV deletions, SV duplications, inversions and translocations) include all SVs identified by a consensus of SV callers, Manta and GRIDSS, regardless of whether they contribute to complex SV events. The only exceptions are in Supp Figure 1C, Supp Figure 3C, Figure 3B and Figure 3C. Here simple SVs are classified separately to clustered SVs and are defined as SVs not belonging clusters of 3 or more SVs as identified by clusterSV (methods detailed in lines 614-618 and the respective figure legends). We have included additional details in the variant calling section of the methods (lines 582-584) to clarify this further.

- 4. Quantification of the burden of DNA rearrangements and copy-number alterations.** The authors need to apply consistent criteria for the classification and quantification of copy-number alterations or rearrangements. For example, in Supp Figure 1A, the median number of duplications is between 10

and 100, but in Supp Figure 1C, the median number of duplications is ~1000. How should one interpret these results?

We apologise for the inconsistent use of log scales in Supp Figure 1. Supp Figure 1A is plotted on a log₁₀ scale whereas Supp Figure 1C uses a natural log transformation. We have remade Supp Figure 1C to also use a log₁₀ scale and to ensure the colour scheme is consistent within the figure. The number of events per boxplot are also slightly lower in Supp Figure 1C as here SVs are additionally classified by size and whether they belong to an SV cluster.

Moreover, different classes of alterations should be quantified separately. For example, gain or loss of a chromosome can arise from a single chromosome missegregation event; the same is true for chromothripsis. In Suppl Figure 2A, many samples have >1,000 “mutational events.” Are the authors counting individual segmental deletions of a chromothripsis pattern as separate events, or grouping them as one instance of chromothripsis?

In Supp Figure 2A, ‘mutational events’ means SNV, CNV or SV variant calls. We have changed the y-axis label to ‘Number of SNVs, SVs and CNAs’ to more accurately reflect this. The burden of clustered/complex events is represented by the number of SVs in SV clusters present per sample. We believe given the necessary uncertainty of assigning segmental consequences to single mutational events such as chromothripsis, this better represents the per sample burden of alteration in complement to Figure 1.

5. **Enrichment analysis.** It is unclear how the authors assess the frequency of a gene or locus being affected by rearrangement. For example, in Figure 2C, the distributions appear to be near identical for many types of rearrangements. Do these distributions simply reflect the distribution of breakpoints that are classified as one or multiple “cSV” clusters?

Our approach to identifying genomic regions enriched for CNAs or SVs is described in the methods section ‘Prediction of CNA/SV mediated driver genes’. In brief, for SVs we use Fishhook to fit a negative binomial regression model on genomic windows to identify regions with a greater density of SVs than expected based on the genomic background rate. For CNAs, we employ the established method GISTIC.

Figure 2C plots the density of SVs and complex SVs by type across chromosome 19. When we exclude SVs contributing to complex SVs from the SV plot we continue to see fairly similar distributions of breakpoints across SV types (below) suggesting that these SV hotspots are not determined solely by the complex SVs captured here (clarified on line 194) but are also not SV type specific.

On excluding complex SV calls that overlap with another complex SV call of a different type, we still observe the peak at *CCNE1* separately for breakage fusion bridges, chromothripsis and pyrgo despite the reduction in power. In addition, complex SVs which are uniquely breakage fusion bridges, or uniquely ecDNA remain significantly enriched on chromosome 19. Uniquely chromothripsis and uniquely chromoplexy calls are also enriched but to a lesser extent and the enrichment is not statistically significant after multiple testing adjustment in this less well-powered dataset. We have added lines 200-204 to the results section to clarify this.

Alternative Figure 2C: The distribution of simple SVs only, on chromosome 19 with the position of *CCNE1* highlighted.

For the enrichment of copy-number alterations, the authors did not distinguish between genes that may be disrupted as passengers. For example, in Figure 3A, disruption of *CEP89* (3Mb from *CCNE1*) appears to follow the same pattern as *CCNE1*; *LEPROTL1* (1Mb from *WRN*) shows a similar concurrence with *WRN*. I tend to think that their co-occurrence is attributed to the positive selection for the disruption of *CCNE1* and/or *WRN*, but not due to *LEPROTL1* or *CEP89* being new drivers.

To distinguish between candidate driver genes and genes disrupted as passengers, we considered both recurrence of CNAs at the gene in question and, by harnessing the power of matched RNAseq, whether CNAs at a gene resulted in significant differential expression. However, this approach does not directly discriminate between candidate genes at the same locus, where they both show differential expression (as is the case with *CCNE1* and *CEP89*). Methods are described in methods sections ‘Identification of genomic regions of recurrent copy number alteration/structural variation’ and ‘Prediction of CNA/SV mediated driver genes’.

We agree that it is likely that disruption of *CEP89* or *LEPROTL1* is a consequence of their close proximity to *CCNE1* and *WRN* respectively and does not suggest that these genes are independent candidate drivers. We have highlighted that these pairs of genes are at the same genomic loci and are likely concurrently impacted by the same SVs and CNAs in lines 252-253 and 267-270. We have now also explicitly stated that we do not believe it likely that they represent independent driver genes (lines 258-260).

Presentation issues:

1. Figure 1.

1.a) Most rearrangements only affect genes that are adjacent to the breakpoints. The crude quantification based on “number of bases disrupted” is misleading.

We disagree that it has been established that rearrangements only affect genes that are adjacent to the breakpoints. Figure 2D demonstrates that genes within the span of SVs or CNAs show differential expression, whether close to the breakpoints or not. In the simplest case of SV

deletions (in the context of the information conveyed in Figure 1), loss of an exon within the span of the deletion would be expected to impact gene function. Additionally, the functional impact of an inversion that spans an exon has not yet been established. Nevertheless, we included Supp Figure 2 to reflect the number of SV calls rather than the extent of genome affected, presenting these data from multiple perspectives.

1.b) Segmental copy-number alterations are accompanied by DNA rearrangements. How did the authors address such redundancy.

There is some redundancy between deletions and duplications identified using SV callers and CNA callers. In common with the rest of the field (e.g. PCAWG) we identified, reported and analysed these events separately, reflecting their different modes of identification and likely differing mechanisms of generation. In general, CNA deletions and duplications are larger reflecting large CNA consequences of chromosomal instability processes that can be detected using lower resolution approaches.

1.c) It would be much more informative to group patients by HRD status/Germline HRD mutations, Stage, and/WGD, and plot the burden of SNVs, SCNAs, and rearrangements that are sorted within each subgroup.

In Supp Figure 8, we group patients according to HRD and WGD status and report the number of SNV, SV and CNA candidate drivers per sample per subgroup. We also present this by candidate driver gene. We also state on lines 297-300 that we do not observe a difference in the number of drivers or the distribution of drivers by mutation type by subgroup.

1.d) The authors need to describe how they identify genes being disrupted by mutations in Fig. 1B and 1C. The suggestion that 84% of all protein coding genes are disrupted by one of four types of alterations is unrealistic.

In Figures 1B and C, 84% of protein coding genes are disrupted by at least one of: a SNV or indel (high/moderate predicted functional impact), CNA or SV spanning at least 1 exon, in at least one of the 324 tumours. We have made it clearer in the figure legend that the first part of Figure 1B is at the cohort level. In a given sample, the proportion of protein coding genes disrupted is much lower than 84%, as shown in the right panel of Figure 1B. We do not suggest that all of these events are drivers and subsequently employ more stringent filtering techniques to identify candidate CNA and SV driver genes (see Figure 3).

2. Figure 2. In Figure 2B, what is being used for clustering? Is it by genes, or regions?

In Figure 2B, samples (columns) are clustered based on the presence of duplications and deletions at CNA hotspots identified by GISTIC analyses presented in Figure 2A. Samples with similar patterns of hotspot activity cluster together and reflect that WGD and HRD samples have differing patterns of hotspot activity.

3. Figure 3. Isn't all the information in C already shown in A?

Figure 3C highlights the percentage of patients carrying alterations of any type at candidate driver genes. In Figure 3A, plotted alterations are filtered by predicted pathogenicity and are not

combined across all alteration types in one plot. In addition, in Figure 3C the impact of clustered SVs versus simple SVs is highlighted supporting the role of complex SVs in amplifying *CCNE1*.

4. Figure 4. A and B. How is the clustering performed? How are the PCs of complex SV signatures determined? C and D. What's new in D that is not contained in C?

We have described our approach in lines 680-683 in the methods section. We have now added more detail clarifying that in our hierarchical clustering approach branches were merged based on Euclidean distances and using complete linkage. Principal component analysis was performed on the prevalence matrix for illustrative purposes to highlight the most important axes of variation in complex SV prevalence.

Figure 4C describes the enrichment or depletion of different types of complex SVs in WGD (green) and HRD (purple) samples whereas Figure 4D highlights the pairwise correlations between all types of complex SVs. These correlations are influenced by the ambiguity of classifying complex rearrangement patterns as single complex SV types which we have highlighted in lines 325-327 of the text.

Interpretation issues/further analysis needed:

5. *CCNE1* is often amplified in HGSOC and tumors with *CCNE1* overexpression show a high prevalence of large tandem duplications (*Cancer Cell* by Menghi et al). The authors should comment on if the same pattern is observed here.

Yes, we do observe the same pattern here. Tumours with *CCNE1* amplification have more ($p = 0.025$) tandem duplications than tumours without *CCNE1* amplification and these duplications are significantly larger ($p < 2.2 \times 10^{-16}$). The median length of duplications in *CCNE1* amplified tumours is ~80kb (consistent with class 2 TDs) in contrast to ~25kb (consistent with class 1 TDs) in tumours without *CCNE1* amplification. We have added lines 196-197 to mention this and cite the study by Menghi et al.

6. How often do complex SVs lead to gene fusions? How does complex SVs or simple SVs alter gene expression after correcting for copy-number changes?

Most algorithms predicting gene fusions are based upon expression data and are notorious for high false positive rates (Haas et al (2019), *Genome Biology*). In spite of this many previous studies of fusion genes have failed to validate such predictions, leading to a literature that is likely to be misleading. We examined the evidence for gene fusions in our HGSOC cohort using a rigorous approach: predicting fusion transcripts based upon the best performing algorithm (STAR-Fusion; Haas et al (2019) *Genome Biology*) and then seeking evidence for a translocation or other rearrangement in the same samples, that might validate those transcripts. We used SV calls from Manta and GRIDSS derived from WGS data, independently of the RNAseq analysis. We found 1757 predicted fusion transcripts across the cohort, but 90% of these were found in only a single sample, representing very rare events in HGSOC or possible false positives. Only 1.4% (25/1757) of fusion transcripts occur in 10 or more samples, and of these, more than half involve exons from two neighbouring genes, potentially reflecting POL2 read-through between those genes. Consistent with this we find that only 0.5% (9/1757) of fusion transcripts coincide with the location of a SV breakpoint in at least one sample. The best supported fusion transcript, joining exons from the *SAA4* and *SAA1* genes, was observed in 49 samples, and in 30 of those samples there was a corresponding SV breakpoint. However, these genes are a duplicated paralogous pair, suggesting

this transcript may be a false positive caused by mismapping of RNAseq reads between closely related genes. We conclude that genuine fusion genes – involving structural variation - are rare in HGSOC and that much of the literature is likely to reflect read-through transcripts encompassing neighbouring genes. We also find that 15% of predicted fusion genes overlap complex SVs, which is not unexpected since many cSVs span many megabases, and only 0.2% (3/1757) overlap a cSV in at least 5 samples. Although we find little convincing evidence for fusion genes in HGSOC, or their involvement in cSVs, we include a new supplementary table (Supp Table 15) and additional methods text (lines 590-593) to describe these data in the hope that they are useful to other researchers.

The field of complex SV prediction has developed relatively recently and the functional impacts of complex SVs, including on tumour expression patterns, are poorly studied. However, even in the case of a single, simple SV it is non-trivial to accurately assess the impact of structural variants on gene expression due to a variety of confounding factors. The technical limitations of tumour derived RNAseq data, sequenced at varying depths across samples, may obscure expression changes for many genes that are not highly expressed. The substantial cellular heterogeneity in HGSOC tumour samples means that the expression levels measured may be influenced by multiple different cell types, including stroma and immune components as well as tumour cells. Also, as the reviewer suggests, the high genome-wide mutational loads observed may also influence expression changes (potentially in cis or in trans) across the genome. These confounding mutations may include SNVs in transcription factors and/or regulatory sites, other co-occurring SVs, CNAs and complex SVs of several different types. In addition, the alterations to mtDNA and the presence of ecDNAs seen in HGSOC may also influence the expression of an unknown number of genes. We are not aware of any gene expression modelling framework capable of accounting for variation in so many confounding variables simultaneously.

An additional issue in assessing the impact of complex SVs on gene expression is sample size, since calculating significant differential expression depends on having enough samples with a gene disrupted by the same complex SV type. Unexpectedly we have found that chromosome 19 is a hotspot for complex SVs, with many overlapping the *CCNE1* locus, which provides us with an opportunity to test the effects of various complex SV types on *CCNE1* expression. We show in Supp Figure 6 that both the copy number and expression of *CCNE1* are often amplified by complex SVs (ecDNA, BFB, chromothripsis, and combinations of all three) beyond the effects of simple SVs at the same locus (Wilcoxon $p = 5.24 \times 10^{-8}$). However, the caveats described in the previous paragraph do – of course - also apply to these estimates of *CCNE1* expression, but we include them as supplementary data for completeness.

7. Do HRD or WGD ovarian cancers follow different evolutionary trajectories? Is there any difference between tumors from patients with germline HRD and those with HRD due to somatic mutations?

We present evidence that HGSOC tumours do indeed follow distinct evolutionary trajectories, largely dictated by the presence of HRD and WGD. This is evident from the patterns of association among complex SVs (Figure 4), such that WGD shows strong co-occurrence with a variety of other complex SVs (particularly chromothripsis, BFBs and ecDNAs). In stark contrast, HRD tumours show a lower incidence of all complex SVs (other than chromoplexy) than expected. The presence of HRD or WGD are also associated with striking differences in the spectrum of CNA seen genome-wide (Figure 2; Supp Figure 2), and in the loads of SNVs seen in mtDNA (Supp Figure 10). However, the

driver genes carried by HRD and WGD tumours are similar (Supp Figure 8), suggesting different mutational paths to impact a common array of driver genes.

We found no evidence for major differences between samples with germline *BRCA1* or *BRCA2* pathogenic ClinVar SNVs (14% of samples) and samples with somatic SNVs or SVs in the same genes (29%). However, this study was not designed to examine differences between samples with HRD derived from germline and somatic variants and is under-powered to address this question.

8. As the authors want to highlight the differences between WGD and HRD ovarian cancers. It would be desirable to present and contrast genomic patterns for WGD and HRD samples separately throughout the study.

In our answer above we describe the major contrasts between the mutational landscapes of tumours with HRD and WGD, and the figures they are presented in, but it is also important to note that these features are not mutually exclusive. Most samples have only WGD (N=81) or only HRD (N=104), but substantial numbers have both HRD and WGD (N=78), and some have neither HRD nor WGD (N=61). As we state in the manuscript, “HRD and WGD are not mutually exclusive but are anti-correlated such that HRD is depleted in samples with WGD (odds ratio (OR) 0.56; Chi-squared $p=0.015$).” Furthermore, the end point for all tumours appears to be similar – in terms of the driver genes altered by the mutations occurring in genomes with HRD or WGD (Supp Figure 8).

9. It would also be broad interest if the authors can compare their findings in ovarian cancers to complex SVs (cSVs) in other types of cancers that show WGD or HRD (e.g., breast cancers).

We have added text to the discussion to discuss the similarities to other tumour types, and to explain the caveats to such comparisons, such as differences in methodologies between studies.

Minor issues:

1. The main tumor genomic features, including CNV load and SV load, are confounded by tumor fraction. Please provide data to rule out this possibility.

Since no individual CNA calling algorithm is perfect, we adopted a conservative approach, only accepting the overlapping CNA calls between three algorithms: CNVkit, CLImAT and PURPLE. Of these algorithms, two out of three make estimates of tumour purity and ploidy: CLImAT (Yu et al (2014), Bioinformatics) and PURPLE (Priestley et al, (2019) Nature) to improve the accuracy of their calls in the presence of widespread aneuploidy in a sample. This conservative approach may result in under-calling of CNA but provides confidence that our consensus CNA calls are robust to variation in tumour sample purity, and also to the frequent whole genome duplication and aneuploidies seen in this cohort.

We also used a conservative, consensus approach for SV calling, combining calls from Manta and GRIDSS. We chose these algorithms as benchmarking has shown they are among the most accurate and consistent SV callers (Cameron et al, (2019) Nature Communications), and they have been shown to maintain accuracy over a range of tumour purities (Cameron et al, (2021) Genome Biology).

2. Please provide details on how the copy number baseline was determined for gains or losses in samples with and without WGD.

See the answer above. Our consensus CNA calls are based upon callers that explicitly account for unusual ploidy seen in tumour samples.

3. Please ensure that the genomic features of each gene are corrected for clonality and that driver genes are involved in clonal events

The focus of this project was to comprehensively describe the mutational landscape of HGSOC for the first time – from SNVs to the most complex rearrangements currently known – and build a catalogue of candidate driver genes. As we show in Figure 3, most alterations to these genes involve CNAs, SVs and complex SVs – on a background of frequent WGD and HRD. Determining the clonality and timing of a range of somatic alterations in complex, highly rearranged genomes is very challenging, and is close to the limits of current methodologies. However, substantial ongoing work aims to address this and will be the subject of a separate manuscript later this year.

Regarding code availability, we have also included a link to the github repository where the primary processing scripts are available:

https://github.com/ailithewing/Structural_variants_BRCA1_2_HRD_inHGSOC.

Reviewer 3

The authors state their impact is in improved understanding of driving events in high-grade serous ovarian cancer: "The driver landscape in HGSOC remains controversial, highlighting the need for comprehensive analyses of the mutational complexity of HGSOC genomes and elucidation of its clinical impact, in larger WGS cohorts"

NOTEWORTHY RESULTS:

**HRD and WGD underlie different evolutionary trajectories to structural diversity

**Deleterious SNV loads in mtDNA are a novel biomarker of poor patient prognosis

**SNVs in mitochondrial genes is higher in tumors with WBG and lower in tumors with HRD - implies less tolerance to mtDNA mutations for HRD deficient tumors.

SIGNIFICANCE:

This work extends the established literature in outlining the molecular events that characterize high-grade serous carcinoma. They point out molecular factors that point to a poorer prognosis.

Are there any flaws in the data analysis, interpretation and conclusions? Do these prohibit publication or require revision? Some imprecisions as noted below.

Is the methodology sound? Does the work meet the expected standards in your field?

YES Is there enough detail provided in the methods for the work to be reproduced? YES

We thank the reviewer for clearly conveying the novel results and overall significance of this study.

LINE 88: most studies would support that the platinum sensitivity of ovarian cancer is quite a bit higher than "62%". You use as your reference ICON8 which was a study of patients randomized to neoadjuvant chemotherapy - these studies tend to enroll patients with a poorer prognosis and likely this impacts the platinum sensitivity response. Anyone who treats ovarian cancer will trip over the "62%" number and think

it inaccurate. I would suggest saying something less definitive like "sensitive to platinum chemotherapy in 70-80% of cases", and quote GOG 111 (McGuire et al, NEJM, 70% RR), GOG 158 (Ozols et al JCO, 88%).

Thank you for your suggestion. We agree with the reviewer that 62% may be an underestimate specific to the ICON8 study and have altered our reported sensitivity rate as suggested. We have quoted the McGuire study with a 73% response rate in the platinum-taxane arm as supporting evidence for the 70-80% range and inserted the less definitive text that the reviewer suggests (line 88). (The GOG158 study has not been cited because that study included only stage III patients whose residual disease was <1cm following surgery so response rate was not measurable by current standards).

PAGE 12 / Figure 4 and associated text. The authors correctly note on lines 169-170 "HRD and WGD are not mutually exclusive but are anti-correlated such that HRD is depleted in samples with WGD (odds ratio (OR) 0.56; Chi-squared $p=0.015$)." However, on page 12 / figure 4 and associated text, they use the terms "strong mutual exclusivity" and "weak mutual exclusivity". This terminology seems imprecise given that mutual exclusivity is a binary condition. I would suggest they standardize their language here and categorize HRD and WGD (and other genomic events that they compare) as "strongly" or "weakly" ANTI-CORRELATED.

We agree that this change in terminology is wise and have altered our language, on lines 322, 325 and in Figure 4, to reflect the negative correlations observed.

LINES 374-396: an assessment of factors impacting OS that concludes that FIGO stage drives prognosis is quite anti-climactic. This study is primarily made up of stage 3/4 tumors. I would be much more interested in a prognosis assessment that REMOVED FIGO stage as a factor (i.e. only assess stage 3C cancers - you have a huge number of them and could make potentially much more interesting associations by eliminating stage as a factor).

Our multivariable analyses adjust for the effect of stage in determining the impact of other factors on overall survival. Nevertheless, we have restricted our analyses to only stage 3 tumours as suggested (Supp Figure 12) and the results are broadly consistent with those from the multivariable analyses of all tumours. We have highlighted this in lines 407-409. The effect of *CEP89/CCNE1* amplification is ameliorated in stage 3 tumours only but this likely reflects the reduction in statistical power.

BRCA germline mutation carriers are known to have a better prognosis. This data is available based on your figure 1. Its not clear to me in figure 6 why germline BRCA was not included in the univariate / multivariate analysis. If you excluded this because of the HRD assessment, then why also include the BRCA SNV status in the analysis?

This is a good point. We have now added in *BRCA1/2* germline status into our survival analyses for consistency. We have remade Figures 6A, B, C and Supp Figure 12 accordingly and the addition of *BRCA1/2* germline status has little impact on our results in addition to the broader HRD term.

Reviewer 4

The authors describe the structural variations and driver genomic events in high-grade serous ovarian cancer. The manuscript is a tour de force for characterizing the structural variants in HGSC and is worth publishing after a few clarifications.

The study is largely descriptive, and most of the impact and scientific novelty lies in their large, unified dataset and descriptions of the mitochondrial mutations. However the methods are rigorous and the manuscript is well-written, figures are clear and comprehensive. Many of the findings are well aligned with previous work. E.g. Previous work has presented a larger (n=510) multi-site WGS dataset and found three divergent mutation-based evolutionary trajectories contributing to survival in HGSC (PMID 37207655). It would be interesting to discuss how these trajectories are distinct from the two trajectories described here.

Lahtinen et al apply medium depth whole genome sequencing (median coverage 36X) to 510 samples from 148 high grade serous ovarian cancer patients. These multi region data provide them with an excellent resource to model clonal evolution and define subgroups based on clonal complexity and divergence. This is beyond the scope of our study. Two of the subgroups defined were enriched for WGD relative to the remaining group. However, copy number profiles and HRD status did not differ between the three groups. Furthermore, Lahtinen et al did not consider SVs in their study so we are unable to determine how complex SV prevalence, a defining feature of one of our trajectories, relates to their subgroups. We have highlighted the potential for incorporating clonal complexity to enhance our understanding of evolutionary trajectories in HGSOC in lines 480-484 of the discussion.

WGD is also a known poor prognostic indicator and has been validated across several studies. This should be mentioned, as well as the papers referred to in the discussion.

We agree that WGD is a known prognostic indicator in the literature. We have included additional references and detail in the paragraph dedicated to WGD in the discussion to highlight this further (lines 455-457). However, interestingly in our high grade serous ovarian specific study we have little evidence that WGD is linked to patient survival. This may be due to the other correlated events (chromothripsis, BFB, ecDNA) that are seen in tumours with WGD, and show contrasting associations with survival (Figure 6A). Some of these events may drive tumour progression in some cases (eg ecDNA, BFB), while others (eg chromothripsis) appear to make tumours vulnerable. We have highlighted this further in the discussion (lines 473-476).

The authors combine published WGS and gene expression data to construct a large WGS HGSOC cohort with matched RNA-seq to date (WGS N=324). I would refrain from superlatives in a scientific paper.

We have altered the statement on lines 129-130 to remove the superlative. Now, 'to construct a large WGS HGSOC cohort with matched RNA-seq (WGS N=324).'

Supplementary Figure 1 has an overlapping graph - and is not evaluable.

We have remade Supplementary Figure 1 and checked all panels are clearly readable.

Dear Reviewers,
We are delighted
that reviewers 3
and 4 have found
no further issues
with our

manuscript “Divergent trajectories to structural diversity impact patient survival in high grade serous ovarian cancer”, and we have made further revisions to address the remaining objections raised by reviewers 1 and 2. Thank you for taking the time to review our manuscript. We have endeavoured to address all additional comments in full as detailed below.

Reviewer #1 (Remarks to the Author):

The authors have adequately addressed my questions regarding the technical aspects of their bioinformatic analysis. However, the authors have not been able to derive much novel insight from the results produced by the bioinformatics analyses. I therefore find the revised manuscript not reaching the level of publication at Nature Communications. My comments below center on the results presented in the main figures.

We believe this study has yielded several novel insights, as follows.

- We show that the genomic chaos seen in these tumours obscures meaningful underlying patterns, namely two divergent evolutionary trajectories, affecting patient survival and causing different genomic aberrations. One involves homologous recombination repair deficiency (HRD) while the other is dominated by whole genome duplication (WGD) with frequent chromothripsis, breakage-fusion-bridges, ecDNA and mutation of mitochondrial DNA. These contrasting evolutionary pathways may also suggest novel therapeutic approaches in HGSOC, since WGD itself has been reported to be a targetable vulnerability (see Discussion).
- These trajectories contribute to structural variation hotspots, containing novel candidate driver genes with significantly altered gene expression. Using a novel approach to use these data to prioritise driver genes we identify *WRN*, a recently proposed pan-cancer tumour suppressor (Shih et al, 2023), that we show is a particularly compelling new therapeutic target in HGSOC.
- We also show that a known HGSOC oncogene, *CCNE1*, coincides with a novel hotspot for multiple complex SV classes. This complexity has important implications for the likely under-reporting of *CCNE1* amplifications in the literature.
- For the first time we show that these heavily disrupted nuclear genomes are in turn associated with alterations to the mitochondrial genome, impacting patient survival. The magnitude of the impact on survival is directly related to the deleteriousness and frequencies of the mitochondrial DNA mutations present, suggesting metabolic perturbations. This new layer of candidate driver mutations carried in mtDNA suggests new directions for research into tumour biology and potential therapeutic targets.
- The frequency of amplification or disruption of known driver genes seen in this cohort is often higher than that seen in prior studies lacking WGS data, emphasising the necessity for whole

Institute of Genetics and Cancer

The University of Edinburgh, Western General Hospital, Crewe Road, Edinburgh EH4 2XU

T: +44 (0)131 651 8500 ed.ac.uk/institute-genetics-cancer @EdinUni_IGC

genome sequencing to adequately characterise structurally diverse tumour types. Our approach therefore also provide a blueprint for studies of other structurally diverse tumour types, including many cancers of unmet need, and demonstrates the potency of new studies exploiting previously published (but often neglected) WGS data.

1. Figure 1A. The authors remain adamant about their way of quantifying the "number of bases disrupted," which I consider to be totally misleading. In Fig. 1A, top, there are several samples with >6,000,000,000 bases disrupted. Given that a diploid human cell contains 6 billion nucleotides from both set of chromosomes, this implies that more than every single nucleotide is disrupted, which is clearly nonsense. This form of presentation reflects a fundamental misunderstanding of genetic alterations. As the authors chose to sort the samples by their arbitrarily defined parameter of "number of bases disrupted," there is no clear pattern between the different types of complex DNA rearrangements, WGD, or HRD status.

We are of the opinion that there are different ways of presenting these data, each of which have advantages and disadvantages. This is why, in the previous version of the manuscript, we quantified the CNA/SV mutation calls in two ways. Firstly we quantified the number of CNA/SV calls per sample (which is dominated by large CNA duplication calls, obscuring other alterations) in the supplementary figures, but also quantified the number of bases disrupted (which makes the full variety of alterations visible) in Figure 1A.

However, we have followed the reviewers preferences and have replaced the top panel of Figure 1A with the previous supplementary figure highlighting the number of SVs and CNAs identified in each tumour instead of the size of genomic regions affected. This removes any ambiguity created by multiple genomic events impacting overlapping regions of the genome which is a common occurrence reflected in the previous figure. As highlighted in the text, Figure 1A is dominated by CNA duplications identified by copy number callers using read depth and variation in allelic frequencies. Figure 1 allows us to visualise the overall landscape of copy number and structural genomic alterations in HGSOc, but we also depict WGD and HRD status, to demonstrate that there are no immediately obvious trends linking complex SVs, WGD, HRD, and the underlying CNA/SV patterns. Although we do not observe a clear relationship between different classes of genomic alteration and WGD or HRD status in this figure, we then go on to employ more sophisticated approaches to interrogate the relationships between these features throughout the manuscript.

2. Figure 1B. It remains unclear how the authors distinguish between genes disrupted by CNA and genes disrupted by structural variants. Segmental copy-number alterations (deletions or duplications) are invariably associated with DNA rearrangements. Are these counted separately, or added together? Judging from the right panel in Fig. 1B, the last row ("any") appeared to be close to the sum of the four numbers above. I consider this highly unlikely, since many rearrangements in ovarian cancers are associated with segmental copy-number alterations (therefore, the genes disrupted by such rearrangements should be classified under the CNA categories.)

In common with others studying WGS, we have used two strategies to flexibly identify a broad spectrum of structural variation and we have discussed the variants identified by each strategy separately throughout. Firstly, we have used somatic CNA (copy number alteration) callers which

are based on changes in read depth and allelic frequencies to identify large deletions and duplications. These changes often affect multimegabase regions of the genome and are therefore efficiently detected using relatively low resolution changes in read depth. These we have termed CNA duplications and CNA deletions throughout for clarity. We have also identified deletions and duplications using SV callers, in addition to identifying copy number neutral SV classes such as inversions and translocations. The SV callers we employ make use of split reads and discordantly mapping read pairs to predict precise breakpoints, and can detect focal events impacting short regions of the genome. Although there is some overlap between some classes of the consensus variants detected by each of these approaches, the abundant technical differences between CNA and SV callers mean that it is non-trivial to robustly integrate all calls from all callers into a unified variant dataset. As a result, we have based our analyses on consensus CNA and SV call sets, similar to the strategy of the landmark, pan-cancer WGS analysis study conducted by ICGC (ICGC/TCGA Pan-Cancer Analysis of Whole Genomes Consortium, Nature, 2020), in which we participated.

In Figure 1B, the 'Any' category reflects the mean number of genes disrupted by at least one type of alteration in a sample. This removes any redundancy between CNA and SV callers identifying the same alteration to a gene in a sample. The sum of the four numbers in Figure 1B (6912.3) is therefore greater than the reported 6009.6. The intention is to provide estimates of the high levels of collateral damage to gene function in these tumours, which are likely to create vulnerabilities.

3. Figure 1C. What insight has been provided by the list of most frequently disrupted genes except for TP53? Most of the genes are not known to contribute to carcinogenesis (including in ovarian cancers). I consider their frequent disruption to reflect their susceptibility to genomic/chromosomal instability. But that result does not need to be highlighted in a main figure panel.

The genes subject to frequent disruption, which may reflect their susceptibility to chromosomal instability, have not previously been highlighted in comprehensive analyses of WGS of a large number of HGSOC. This figure further highlights that recurrent gene disruption by SNV/indel is rare beyond *TP53* and that the mutational landscape of HGSOC is dominated by CNAs and SVs, which result in a large amount of collateral damage to protein-coding genes (Figure 1B). However, in Figure 1C we demonstrate that the most frequently disrupted genes are not enriched for known cancer genes. This is important given the large cancer genomics literature, which contains many studies that simply report the most frequently disrupted genes as new insights into tumour biology. We agree with the reviewer that their frequent disruption is likely to reflect their susceptibility to alteration, as we make clear in the legend to Figure 1. Indeed, this is the rationale for highlighting these genes in panel Figure 1C.

4. Figure 2B. This is in my view one of a few informative figure panels in this study. What are the duplication and deletion hotspots being used for clustering? And What's the rationale for performing a clustering based on such hotspots? Judging from the result, I see that HRD samples show more deletions and WGD samples show more duplications. This pattern may be even more clear if the authors simply group samples based on HRD/WGD status, e.g., HRD (no WGD), HRD & WGD, and WGD (no HRD), and then sort by CNA burdens per sample.

The rationale for this clustering was to demonstrate the influence of the overall genomic state in a tumour (reflected in HRD and WGD) on the burdens of CNA observed at hotspots, as described in the legend of Figure 2. The duplication and deletion hotspots were identified as described in the relevant methods section 'Prediction of CNA/SV mediated driver genes'. In short, we identified hotspots of CNA duplication and CNA deletion using the established method GISTIC2 applied to the consensus CNA calls. The presence of CNA deletions or CNA duplications at these hotspots was then used to cluster samples with similar patterns of affected CNA hotspots. We do report an increased rate of duplications in WGD samples and deletions in HRD samples in lines 170-171 of the text and in Supp Figures 2B and C. We have also included Supp Figure 2A which is Figure 2B with samples ordered by HRD/WGD status and CNA hotspot activity as suggested, which again reflects this relationship.

5. Figure 2C. Is *CCNE1* a deletion hotspot, duplication hotspot, or a SV hotspot? Are the authors referring to "deletion-type" or "duplication-type" SVs when they use "Deletion" or "Duplication"? There has been much confusion about these terminologies in the literature. I suggest the authors stay away from "deletion SV" or a "deletion type SV" but focus on relative copy-number status (i.e., gene copy number relative to the genome-wide average), and rearrangement breakpoints (regardless of their type or orientation).

CCNE1 lies in SV duplication, SV deletion and inversion hotspots, which were separately identified using the fishhook algorithm (detailed in methods). This may seem confusing, but it is clear in Figure 2C that the burdens of multiple classes of SV peak at *CCNE1* on chromosome 19 and that this is likely to be due to the enrichment of complex SVs (such as BFBs, ecDNAs and chromothripsis) on this chromosome. As clarified above in response to point 2, SV deletion refers to typically smaller deletions identified by SV callers applied to paired reads rather than CNA deletions identified using changes in read depth. We discuss SV calls (from SV callers) precisely so that we can clearly distinguish them from the technically distinct CNA calls (from CNA callers), in common with many other authors.

6. Figure 2D. This figure panel is confusing for a variety of reasons. First, the use of SV deletion and CNA deletion etc. is confusion. Second, the meaning of the color of the labelled genes is not explained. Why are there two *CCNE1* labels? Finally, the authors should annotate which genes have been implicated in ovarian cancer oncogenesis and which genes are likely affected because of their proximity to cancer genes.

As described in the figure panel and in methods, this panel shows the impact on gene expression for genes within CNA or SV hotspots. In Figure 2D, there are two *CCNE1* labels in different colours to reflect two of the types of hotspot that *CCNE1* lies in, SV duplication and inversion hotspots, as described above. The significant expression changes reported are seen in the presence of SV duplication and inversion respectively. The meaning of the colours of each are stated in the associated legend on the right of Figure 2D. As discussed above, SV deletions are deletions identified by SV callers and CNA deletions are those identified by CNA callers. We discuss the genes that have (or have not) been implicated in HGSOC in the relevant text section: 'Hotspots of structural alteration implicate novel candidate driver genes'.

7. Figure 3. This figure is essentially a re-cap of Figure 1. Despite the authors acknowledging that many of the genes listed in panel A are likely passengers, they still label them as "candidate driver genes." I consider this to be very misleading.

As mentioned above, Figure 1 depicts the overall mutational landscape of the HGSOC tumour genome, including genome-wide burdens, the high levels of complex SVs, and the extent of collateral damage to known protein-coding genes (ie all genes). In contrast to Figure 1, the variants in Figure 3A have undergone substantial filtering to depict only variants in rigorously predicted driver genes, using the computational predictions from WGS and orthogonal RNAseq data detailed in the text. The genes included in this figure ('Diverse somatic mutation classes underlie HGSOC candidate driver genes') are candidate driver genes yielding significant results from SNV-based driver prediction algorithms (as detailed in methods), and those within significant CNA or SV hotspots that also show significant differential expression (methods). The CNA and SV candidate driver genes that emerge in Figure 3 are therefore required to satisfy multiple forms of evidence. They must be genes with a known role in tumourigenesis, (ie included in the cancer gene census), they must be recurrently altered at a significant CNA/SV hotspot, and they must be significantly differentially expressed in response to the candidate driver events at the hotspot. To our knowledge, these criteria are more stringent than previous studies of driver genes mediated by structural alterations, and the set of candidate driver genes presented is the most comprehensive to date for HGSOC. Certain genes, such as *LEPROTL1* and *CEP89*, meet all of the stringent criteria for candidacy and therefore have been included in this figure for a complete and accurate representation of the data. However, we accept and highlight clearly that as they occur in the same hotspots as *WRN* and *CCNE1*, based on our current knowledge of HGSOC drivers, they are unlikely to be the main driver genes at these loci and may be passengers. The data at the moment is not sufficiently powered to rationally exclude such genes. It is also possible that multiple driver genes are simultaneously altered by CNA/SV variants, as has been reported elsewhere in other tumour types.

8. Figure 4. This figure can be potentially interesting. Figure 4A is essentially a repeat of the content in Fig. 1A. The meaning of the components in Fig. 4B is unclear to me. Fig. 4C and Fig. 4D are interesting and will benefit from being presented together. In Fig. 4E, it will be more clear to simply compare genomes with or without WGD, instead of relying on the % of genome duplicated as an indirect measure.

As mentioned above, Figure 1 depicts the overall mutational landscape of the HGSOC tumour genome, and does not examine the co-occurrence of complex SVs, HRD and WGD, or address the question of whether any patterns of co-occurrence are significant. In contrast, the content in Figure 4A provides a detailed (and entirely novel) account of the high prevalence and variety of complex SVs in HGSOC, as well as the relationships between different classes of complex SVs across this large WGS cohort. Rows are clustered based on similarity in sample prevalences between complex SV types which is not shown in Figure 1, and in Figure 4B PCA is employed as an alternative approach to depict essentially the same relationships between these types, HRD and WGD. On reflection, we have removed Figure 4B to instead focus on the strength of the associations

between these entities. The new Figure 4B reports the precise enrichments of each complex SV type in samples with (versus those without) WGD and in samples with (versus those without) HRD.

In Figure 4E (now Figure 4D), we have considered the percentage of the genome duplicated to present a more nuanced picture of the relationship between increased genomic content and complex SVs. The definition of WGD in the literature usually relies on hard thresholding of this measure at 50% for example, though it is known that WGD events often degrade via large deletions, and it is unlikely that a single percentage threshold will capture this variation. The flexible representation in Figure 4D, across a range of possible thresholds, highlights that the relationship between complex SVs and WGD is robust.

9. Figure 5. I cannot comment on this as I do not know about mitochondrial genetics.

This Figure presents the first comprehensive study of somatic mtDNA variants in HGSOc. It also demonstrates the unexpected (and novel) association of mtDNA variants that disrupt mitochondrial gene function with patient survival, and shows that this association is modulated by heteroplasmy (the frequencies of such variants in the mitochondrial population).

10. Figure 6. I consider this figure to be the most novel result of the paper. It will be further improved if the authors eliminate passenger genes (i.e., those in close proximity to a known cancer driver) from the regression analysis. Panel D is a very clean result (although I'm unsure if similar results were reported previously). In Panel F, can the authors clarify their definition of "severe chromothripsis"? Is it based on the # of chromosomes inferred to have undergone chromothripsis, or the total number of breakpoints in chromothriptic rearrangements?

As mentioned above (point 7) all of the candidate driver genes included in the modelling are supported by multiple lines of evidence, and there is no rational basis for excluding them. However, we have re-fitted the multivariable model excluding events at the *LEPROTL1* and *CEPB9* genes and the amended forest plot has been added to Supp Figure 12. However, the exclusion of these genes has little effect on the regression results. Our estimate of the effect of *WRN* CNA deletion on overall survival increases slightly, as expected. However the removal of *CEPB9* SV duplications from the model, means we now fail to detect the impact of disruption of the complex *CCNE1/CEPB9* locus on overall survival. We suspect that this is because some of the SV duplication calls only encompass *CEPB9*, but in reality they may functionally disrupt both genes. Thus SV duplication calls intersecting either *CEPB9* or *CCNE1* may be markers of complex SVs across the locus, encompassing both genes.

Severe chromothripsis is defined on line 338 of the manuscript and now additionally in the Figure 6 legend, as the presence of more than 2 chromosomes inferred to have undergone chromothripsis in a tumour.

We are happy to hear that the latest reviewer, Reviewer #5, finds that we have reasonably

addressed the comments of the previous reviewers. We have also endeavoured to address the additional issues raised by Reviewer #5 in full below (our replies in blue).

Reviewer #5

One major limitation of the study, which I don't know if came up during earlier rounds of review, is that the study under-utilizes the RNAseq data. The authors only use these data for the prioritization of candidate drivers, but it can be used in multiple additional and interesting ways. Two prominent examples:

- For novel candidate driver genes, is there any evidence that specific pathways in which they participate are dysregulated in the tumors that carry these genetic alterations?

The issue of the potential for RNAseq analysis has indeed been raised in previous rounds of reviews. The novel candidate driver genes we have identified are mediated by forms of structural aberrations such as structural variants (SVs) and copy number alterations (CNAs), and often involve complex combinations of SV/CNA. However, even in the case of a single, simple SV it can be non-trivial to accurately assess the impact on gene expression, due to a variety of confounding factors. The technical limitations of tumour derived RNAseq data, sequenced at varying depths across samples, may obscure expression changes for many genes that are not highly expressed. The substantial cellular heterogeneity in HGSOC tumour samples means that the expression levels measured may be influenced by multiple different cell types, including stroma and immune components as well as tumour cells. In addition, the high genome-wide mutational loads observed in HGSOC samples, mean that every sample carrying an SV/CNA of interest will also carry many other mutations that may also influence expression changes (potentially in cis or in trans) across the genome. These confounding mutations may include SNVs in transcription factors and/or regulatory sites, other co-occurring SVs, CNAs and complex SVs of several different types. These confounding factors mean that there are many caveats associated with any dysregulated genes or pathways observed in our tumour cohort, and we had therefore avoided reporting such results.

However, for completeness and as the reviewer suggests, we now report all significantly enriched pathways (as GO and KEGG terms) for the differentially expressed genes associated with all driver candidates (Supplementary Tables S16 and S17). The analysis shows some expected results among the strongest enrichments, such as increased ribosome biogenesis associated with PTEN or RB1 deletion (Supp Table S17), but suffers the caveats discussed above, and is therefore challenging to interpret.

We have also added new text to the revised manuscript (p8) describing these data:

“Differentially expressed gene functions and pathways associated with the presence of candidate CNA/SV driver genes are detectable (Supplementary Tables S16 and S17), but should be interpreted with caution, due to the substantial cellular heterogeneity and potential presence of confounding mutations in HGSOC tumour samples.”

Institute of Genetics and Cancer

The University of Edinburgh, Western General Hospital, Crewe Road, Edinburgh EH4 2XU

T: +44 (0)131 651 8500 ed.ac.uk/institute-genetics-cancer @EdinUni_IGC

- As mentioned above, it could be telling to examine mitochondria-related gene expression in the context of the mtDNA mutations.

There are similar caveats to those discussed above for studies of differential expression between samples with and without the disruptive mtDNA mutations we have discovered to be associated with poorer survival. However, as suggested, we have examined mitochondrial and nuclear gene expression between tumour groups with and without these mtDNA variants. The results are presented in a new Supplementary Figure 13 (also included below), and show modest differences, with only 47 nuclear genes significantly differentially expressed (DE). Of these, only CYP24A1, linked to vitamin D metabolism, is known to have a function in mitochondria. However, GSEA gene set analysis of the DE genes did identify enrichment of pathways related to mitochondrial function in wildtype tumours (lacking mtDNA mutations), notably OXPHOS and reactive oxygen species production. This implies, as expected, that disruptive mtDNA variants may reduce the activity of some mitochondrial pathways.

We have added new text to the revised manuscript (p13) describing these data:

“Comparisons of gene expression between tumour groups with and without deleterious mtDNA variants show modest differences, with only 47 nuclear genes significantly differentially expressed (Supp Figure S13A). Pathways enriched in these genes related to higher OXPHOS and reactive oxygen species production in tumours lacking variants suggesting, as expected, that disruptive mtDNA variants may compromise normal mitochondrial function (Supp Figure S13B).”

Supp Figure 13: Pathogenic mtDNA variants are associated with differential expression. (A) Volcano plot depicting the differential expression between HGSOC tumour samples with and without pathogenic mtDNA variants, indicating (blue points) only 47 significantly differentially expressed (DE) genes across the mitochondrial and nuclear genomes. (B) Gene set enrichment analysis using the GSEA tool comparing the ranked genes of tumours with and without pathogenic mtDNA variants, reveals pathways enriched ($p < 0.001$) in each group of tumours. Genes were ranked based on the DESeq2 test statistic and GSEA was

performed using the fGSEA R package (v1.16.0) with a minimum gene set size of 10, a maximum of 500 genes, and 100,000 permutations, against the MSigDB Hallmark gene set collection.